# Using CALIOP to constrain blowing snow emissions of sea salt aerosols over Arctic and Antarctic sea ice

Jiayue Huang[1], Lyatt Jaeglé[1], Viral Shah[1]

[1]Department of Atmospheric Sciences, University of Washington, Seattle, WA

*Correspondence to*: jaegle@uw.edu

**Abstract.** Sea salt aerosols (SSA) produced on sea ice surfaces by blowing snow events or lifting of frost flower crystals have been suggested as important sources of SSA during winter over polar regions. The magnitude and relative contribution of blowing snow and frost flower SSA sources, however, remain uncertain. In this study, we use 2007–2009 aerosol extinction coefficients from the Cloud-Aerosol Lidar with Orthogonal Polarization (CALIOP) instrument onboard the Cloud-Aerosol

Lidar and Infrared Pathfinder Satellite Observation (CALIPSO) satellite and the GEOS-Chem global chemical transport model to constrain sources of SSA over Arctic and Antarctic sea ice. CALIOP retrievals show elevated levels of aerosol extinction coefficients (10–20 Mm$^{-1}$) in the lower troposphere (0–2 km) over polar regions during cold months. The standard GEOS-Chem model underestimates the CALIOP extinction coefficients by 50–70%. Adding frost flower emissions of SSA fails to explain the CALIOP observations. With blowing snow SSA emissions, the model captures the overall spatial and seasonal

variation of CALIOP aerosol extinction coefficients over the polar regions, but underestimates aerosol extinctions over Arctic sea ice in fall–early winter and overestimates winter-spring extinctions over Antarctic sea ice. We infer the monthly surface snow salinity on first-year sea ice required to minimize the discrepancy between CALIOP extinction coefficients and the GEOS-Chem simulation. The empirically-derived snow salinity shows a decreasing trend between fall and spring. The optimized blowing snow model with inferred snow salinities generally agrees with CALIOP extinction coefficient to within

10% over sea ice, but underestimates them over the regions where frost flowers are expected to have a large influence. Frost flowers could thus contribute indirectly to SSA production by increasing the local surface snow salinity and, therefore, the SSA production from blowing snow. We carry out a case study of an Arctic blowing snow SSA feature predicted by GEOS-Chem and sampled by CALIOP. Using backtrajectories, we link this feature to a blowing snow event which occurred 2 days earlier over first-year sea ice and was also detected by CALIOP.

**1 Introduction**

Sea salt aerosols (SSA) are produced via wave breaking in the open ocean (Lewis and Schwartz, 2004; De Leeuw et al., 2011 and references therein). Over polar regions, SSA can also be generated via sublimation of saline blowing snow (Simpson et al., 2007; Yang et al., 2008), wind-blown frost flower crystals (Rankin et al. 2000; Domine et al., 2004; Xu et al. 2013), and by leads in sea ice (Nilsson et al., 2001; May et al, 2016). SSA production from blowing snow events requires strong winds (>

7 m/s) and depends on the salinity of snow cover on sea ice (Yang et al., 2008). Frost flowers grow over new sea ice formed

from open leads under low ambient temperature ($< -20^{\circ}$C) (Kaleschke et al., 2004). These two sea ice sources of SSA have been proposed to help explain polar observations of wintertime maxima in SSA mass concentrations (Wagenbach et al., 1998; Weller et al., 2008; Jourdain et al., 2008; Udisti et al., 2012; Huang and Jaeglé, 2017), the depletion of sulfate-to-sodium mass ratio in winter SSA relative to bulk sea water at sites in Antarctica (Wagenbach et al., 1998; Rankin et al., 2000; Jourdain et

al., 2008; Hara et al., 2012) and some sites in the Arctic (Jacobi et al., 2012; Seguin et al., 2014), as well as the increase in $Na^{+}$ deposition fluxes during glacial periods relative to interglacial periods (Wolff et al. 2006; Fischer et al., 2007; Abram et al., 2013).

In a previous study (Huang and Jaeglé, 2017), we used the GEOS-Chem chemical transport model to examine the relative roles of blowing snow and frost flowers as sources of polar SSA during winter. Our study was based on the blowing snow

parameterization developed by Yang et al. (2008) and the frost flower parameterization of Xu et al. (2013). We compared our simulations to in situ observations of SSA mass concentrations at three surface sites in the Arctic and two sites in coastal Antarctica, showing that blowing snow appeared to be the dominant source of polar SSA during winter. Here, we further constrain the spatial and temporal distribution of polar sources of SSA by using observations of aerosol extinction coefficients from the Cloud-Aerosol Lidar with Orthogonal Polarization (CALIOP) instrument onboard the Cloud-Aerosol Lidar and

Infrared Pathfinder Satellite Observations (CALIPSO) satellite together with the GEOS-Chem model.

One of the main uncertainties in estimating blowing snow emissions is the salinity of snow on sea ice. In our previous work (Huang and Jaeglé, 2017), we assumed constant surface snow salinity over Arctic (0.1 practical salinity unit, or psu) and Antarctic (0.03 psu) sea ice. In reality, surface snow salinity is highly variable in time and space. The sources of sea salt in snow over sea ice include upward migration of brine from the sea ice surface, incorporation of frost flowers and SSA deposition

from the nearby open ocean (Domine et al., 2004). Initial sea ice formation is accompanied by upward salt transport, such that first-year sea ice (FYI) has a high salinity reaching 20–100 psu at the sea ice surface (Weeks and Lee, 1958; Martin, 1979; Weeks and Ackley, 1986). Nakawo and Sinha (1981) found that sea ice salinity decreases rapidly within the first week of sea ice formation in fall, and then decreases more slowly between December and May in the Canadian Arctic. Worby et al. (1998) showed that ice with a thickness of less than 0.05 m displayed salinities of 9–28 psu, while ice thicker than 0.05 m had salinities

of 4–8 psu, which decreased linearly with ice thickness. As snow accumulates on FYI throughout winter, the brine is wicked upward resulting in brine-wetted snow (Barber et al., 1995). Snow salinity is highest in the first 10 cm above FYI, with values of 1–20 psu (Geldsetzer et al., 2009) and then decreases rapidly as the snow cover gets thicker, with low salinities on the surface of thick snow (Nandan et al., 2017). Older and thicker multiyear sea ice (MYI), is desalinated by flushing and gravity drainage during repeated summer melt cycles, such that the overlaying snow has very low salinity. Krnavek et al. (2012)

reported that surface snow salinity sampled on MYI was 0.01 psu, compared to 0.1 psu for snow on thick FYI and 0.8 psu on recently frozen thin FYI in March near Barrow, Alaska.

The role of frost flowers as a direct source of SSA remains subject to debate. Some studies have shown that strong winds inhibit frost flower formation and bury existing frost flowers with snow (Perovich and Richeter-Menge, 1994; Rankin et al., 2000), while field experiments show that frost flowers are difficult to break (Domine et al., 2005; Alvarez-Avilez et al., 2008; Obbard et al. 2009). In addition, laboratory experiments show that evaporating frost flowers form a cohesive chunk of salt (Yang et al., 2017), which is unlikely to be a direct source of SSA even when exposed to large wind speeds (Roscoe et al., 2011).

In this study, we use 3 years (2007–2009) of CALIOP aerosol extinction coefficients to constrain the spatial and temporal distribution of polar SSA emissions with the GEOS-Chem model. Satellite observations and GEOS-Chem simulations are described in section 2. In section 3, we evaluate the model's ability to reproduce observed aerosol extinctions over the Arctic and Antarctic sea ice regions with and without sea ice sources of SSA. In section 4, we develop an empirical parameterization of seasonally-varying surface snow salinity on FYI. In section 5, we conduct a case study of an Arctic blowing snow event and the resulting SSA observed by CALIOP.

## 2 Observations and model simulations

### 2.1 CALIOP observations of aerosol extinction coefficients

The CALIOP Lidar measures backscatter signals of optical pulses at 532 and 1064 nm (Winker et al., 2009). CALIOP samples the optical properties of clouds and aerosols during daytime and nighttime with a 16-day repeat cycle. It has a sampling resolution of 335 m horizontally and 30 m vertically below 30–40 km altitude. In this study, we use vertical profiles of 532 nm aerosol extinctions from CALIOP Level 2 (L2) version 4.10 profile data for 2007–2009 (Winker, 2016). L2 data is retrieved with a set of algorithms, which identify the cloud and aerosol layers and classify their feature types (Liu et al., 2009). Extinction-to-backscatter ratios (lidar ratios) are assigned based on the aerosol types for calculations of L2 aerosol extinctions (Omar et al, 2009). Tesche et al. (2014) showed that CALIOP v3.01 aerosol extinctions coefficients overestimated in situ observations in the Arctic by 47%. For version 4.10 product, a new subtype "dusty marine" is assigned when dust and marine aerosol coexists, with a lidar ratio 33% smaller than that of polluted dust (Winker, 2016), which brings the in situ and CALIOP extinction coefficients in closer agreement (not shown). The layer detection algorithm is performed downward for single shots and profiles are averaged horizontally at 1, 5, 20 and 80 km to achieve good signal-to-noise-ratio (SNR) for aerosol retrievals (Winker et al., 2009). The estimated detection sensitivity of the CALIOP 532-nm channel varies with different horizontal averaging, with better sensitivity at larger horizontal averaging (80 km). Nighttime CALIOP extinction coefficients have better sensitivity than daytime observations, which are affected by noise from solar radiation scattering (Winker et al., 2009).

We average monthly CALIOP L2 aerosol extinction profiles poleward of 60º over a 2° latitude by 5° longitude horizontal grid in 60 m vertical bins, using the same approach as Winker et al. (2013). When no aerosols are detected and the layer is classified

as "clear air", we assign it an extinction coefficient value of 0.0 km$^{-1}$. Following Winker et al. (2013), we exclude the following aerosol layers from our gridded averages: (1) all aerosol layers within 60 m of the surface to avoid surface contamination, (2) layers with a Cloud Aerosol Discrimination (CAD) score falling outside the range of −100 to −20, (3) aerosol layers with uncertainty flags of 99.9 km$^{-1}$ and the layers beneath, (4) extinction QC flags indicating possible large errors, (5) "clear air" under the lowest detected aerosol layer with base below 250 m to avoid low bias for undetected surface-attached aerosol. In addition, we exclude very high values of aerosol extinctions (> 0.35 km$^{-1}$) below 2 km poleward of 60° during cold months (September–May for the Arctic and March–November for the Antarctic), as they are likely related to diamond dust misclassified as aerosol (Di Pierro et al., 2013).

The CALIOP nighttime retrievals over polar regions are limited during summer months (May–July in the Arctic, November–January in the Antarctic), with maximum latitudinal extents of 55° latitude beyond which only daytime retrievals are available. Daytime retrievals have higher detection thresholds and can only detect aerosol layers with relatively high extinctions (Fig. S1–2). Therefore, at a given latitude fewer aerosol layers are detected in the daytime retrievals, and the average daytime extinction coefficients are lower than the nighttime ones over polar regions (Fig. S3). Most of our work is based on analysis of nighttime CALIOP retrievals during winter over polar regions. However, to reconstruct the full seasonal cycle of aerosol extinctions over polar regions, we calculate nighttime equivalent aerosol extinction profiles by combining both daytime and nighttime CALIOP extinctions coefficients, following the algorithm described in Di Pierro et al. (2013). This approach provides an empirical correction for the differences in detection sensitivity and aerosol extinctions in the daytime CALIOP retrievals (more details are given in the Supplement).

## 2.2 The GEOS-Chem chemical transport model

We use the GEOS-Chem (v10-01) 3-D global chemical transport model (Bey et al., 2001) driven by meteorological fields from the Modern Era Retrospective-Analysis for Research and Applications (MERRA; Rienecker et al., 2011). The MERRA assimilated meteorological fields have a native horizontal resolution of 0.5° latitude by 0.666° longitude with 72 vertical levels, which we regrid to 2°×2.5° horizontal resolution and 47 vertical levels with merged levels above 80 hPa.

We conduct a three-year (2007–2009) global simulation of tropospheric aerosol-oxidant chemistry. The model is initialized with a 1-year spin up. Global anthropogenic emissions are from EDGAR v4.2 (Emissions Database for Global Atmospheric Research, Olivier and Berdowski, 2001) for 1970–2008. For the years after 2008, anthropogenic emissions are scaled relative to year 2008, based on government statistics for different countries/regions (van Donkelaar et al., 2008). Over North America, the anthropogenic emissions are from the 2011 National Emissions Inventory (NEI2011; https://www.epa.gov/air-emissions-inventories/2011-national-emissions-inventory-nei-data) produced by the US Environmental Protection Agency (EPA), with

annual scaling factors from the EPA for other years. Over Asia, the anthropogenic emissions are from the MIX emission inventory (Li et al., 2017). Over Europe, we use the Co-operative programme for monitoring and evaluation of long range transmission of air pollutants in Europe (EMEP) anthropogenic emissions (http://www.ceip.at/review-of-inventories/). Monthly biomass burning emissions are from the Global Fire Emissions Database version 4 (GFEDv4, van der Werf et al.,
2010). Black carbon (BC) and organic carbon (OC) emissions are based on the Bond et al. (2007) monthly emission inventory, including sources from fossil fuel and biofuel. Dust emissions are based on the dust entrainment and deposition scheme from Zender et al. (2003). Biogenic emissions of volatile organic compounds (VOCs) are from the Model of Emissions of Gases and Aerosols from Nature version 2.1 (MEGAN 2.1, Guenther et al., 2012). The $HO_x$-$NO_x$-VOC-$O_3$-$BrO_x$ tropospheric chemistry chemical mechanism is described in Mao et al. (2010, 2013) with recent updates in biogenic VOC chemistry (Fisher
et al., 2016; Travis et al., 2016). The gas-particle partitioning of $SO_4^{2-}$-$NO_3^-$-$NH_4^+$ aerosol is computed with the ISORROPIA II thermodynamic module (Fountoukis and Nenes, 2007), as implemented by Pye et al. (2009).

The open-ocean emissions of SSA are a function of wind speed and sea surface temperature (SST) as described in Jaeglé et al. (2011). In Huang and Jaeglé (2017), we inferred that wave-breaking SSA emissions are suppressed during summer at coastal polar sites with cold waters (SST < 5°C). As in Huang and Jaeglé (2017), we reduce SSA emissions for these cold waters. We
use two SSA size bins: accumulation mode ($r_{dry}$ = 0.01−0.5 μm) and coarse mode ($r_{dry}$ = 0.5–8 μm).

Advection is based on the Lin and Rood (1996) advection algorithm and boundary layer mixing is computed using the non-local scheme in Lin and McElroy (2011). Dry deposition in the GEOS-Chem follows a standard resistance-in-series scheme based on Wesely (1989) as described by Wang et al. (1998). Dry deposition of SSA in the model follows the Zhang et al. (2001) size-dependent scheme over land, and is calculated based on the Slinn and Slinn (1980) deposition model over ocean
and sea ice, as implemented by Jaeglé et al. (2011) in GEOS-Chem. The strong size-dependence of SSA deposition is taken into account by integrating the dry deposition velocity over each of the 2 SSA size bins using a bimodal size distribution including growth as a function of local relative humidity (RH). The hygroscopic growth of SSA follows the parameterization of Lewis and Schwartz (2006). Sedimentation of SSA is calculated throughout the atmospheric column based on the Stokes velocity scheme. Wet deposition of aerosol includes convective updraft, washout and rainout from precipitation (Liu et al.,
2001), as well as snow scavenging (Wang et al., 2011). The aerosol extinction coefficients at 550 nm calculated in GEOS-Chem are a function the mass concentrations, extinction efficiency and mass density based on Mie theory, and take into account the hygroscopic growth of aerosols as described in Martin et al. (2003), with an updated size distribution for SSA (Jaeglé et al., 2011).

The blowing snow SSA emissions in GEOS-Chem are based on the parameterization of Yang et al. (2008, 2010) as
implemented by Huang and Jaeglé (2017). The SSA production from blowing snow is a function of RH, temperature, age of snow, snow salinity, and wind speed. The size distribution of wind-lifted snow particles follows a two-parameter gamma distribution (Yang et al., 2008 and references therein). Once sublimated, snow particles are released as SSA particles. We

assume that 5 SSA particles are produced per snowflake (N=5) based on a comparison against observations of submicron SSA mass concentrations at Barrow, Alaska (Huang and Jaeglé, 2017). The size distribution of blowing snow SSA is determined from the size distribution of snow particles, N, and salinity. The resulting emitted mass of blowing snow SSA is obtained by integrating this size distribution into the two SSA size bins. In our previous work, we had assumed a uniform salinity on Arctic

(0.1 psu) and Antarctic sea ice (0.03 psu) based on mean observations of surface snow salinity (Mundy et al., 2005; Krnavek et al., 2012). Here we use these salinities for FYI, but now take into account the lower surface snow salinity of older sea ice, by assuming that MYI snow salinity is 10 times lower than on FYI (Krnavek et al., 2012): 0.01 psu on Arctic MYI snow and 0.003 psu on Antarctic MYI snow.  We calculate a mean snow age of 3 days for the Arctic and 1.5 days for the Antarctic from MERRA meteorological fields.

Frost flower SSA emissions follow the emission scheme of Xu et al. (2013), which is based on the empirical wind-dependence of Shaw et al. (2010) and the potential frost flower (PFF) coverage of Kaleschke et al. (2004). The PFF is a function of ambient air temperature, and frost flowers are formed on very new and young sea ice once the ambient air temperature is cold enough (< about –20°C). We set a threshold of 10 cm for the thickness of newly formed sea ice beyond which we assume that frost flowers do not form due to inefficient brine transport through thicker sea ice. The size distribution of SSA from frost flowers

follows a lognormal size distribution with a geometric mean diameter of 0.015 μm and a geometric standard deviation of 1.9 (Xu et al., 2013). This size distribution is integrated into the two GEOS-Chem SSA size bins to obtain the emitted mass of SSA from frost flowers.

The sea ice concentration boundary conditions in MERRA are derived from the weekly product of Reynolds et al. (2002), which is based on Special Sensor Microwave Imager (SSMI) instruments on Defense Meteorological Satellite Program

(DMSP) satellites. The weekly products have an original spatial resolution of 1°×1°, and are linearly interpolated in time to each model time step. For each year, we use the preceding summertime minimum sea ice extent in MERRA (September in the Arctic and February in the Antarctic) to infer the location of MYI. FYI extent is calculated by subtracting MYI extent from total sea ice extent (Fig. S4).

In this study, we neglect the role of leads as a source of SSA as we found in Huang and Jaeglé (2017) that while this additional

source could potentially be important on local scales near leads, overall the regional increase in SSA emissions is less than 10%.

Our standard simulation (STD) includes tropospheric aerosol-oxidant chemistry and SSA emissions from the open ocean. The STD+Snow simulation is the STD simulation to which we add SSA emissions from blowing snow as in Huang and Jaeglé (2017), with surface snow salinities as described above (0.1 psu on FYI and 0.01 psu on MYI over the Arctic, and 0.03 psu on

FYI and 0.03 psu on MYI over the Antarctic). The STD+FF simulation is the STD simulation with SSA emissions from frost

flowers. In Section 4, we develop an optimized blowing snow simulation (STD+Opt. Snow), with seasonally varying snow salinity on FYI.

The GEOS-Chem simulations are sampled at the time and location of the CALIOP overpasses and averaged over the same horizontal and vertical grid. For comparison to CALIOP observations, we apply the CALIOP nighttime detection threshold to the model, setting the modeled backscatter coefficients to 0 Mm$^{-1}$ sr$^{-1}$ for backscatter values lower than 0.2 Mm$^{-1}$ sr$^{-1}$.

## 3 Model evaluation with CALIOP observations

### 3.1 Arctic

The Arctic cold season (November–April) CALIOP extinction coefficients in the lower troposphere (0–2 km above sea level, asl) display values of 10–30 Mm$^{-1}$ (Fig. 1a). The largest extinction coefficients occur over the open-ocean regions of the Greenland and Barents Seas. In addition, significant aerosol extinction coefficients (10–20 Mm$^{-1}$) are seen over the sea ice covered Chukchi Sea, East Siberian Ocean, Laptev Sea, Kara Sea and Canadian Arctic Archipelago (Fig. 1a). While the STD GEOS-Chem simulation reproduces the pattern of extinctions over the open ocean regions, it fails to capture the enhancements over sea ice, underestimating aerosol extinction coefficients by ~10 Mm$^{-1}$ over the central Arctic (Fig. 1b). The normalized mean bias (NMB=100×($\overline{Model}/\overline{Obs}$ − 1), with $\overline{Model}$ and $\overline{Obs}$ representing mean observed and modeled values) is –55% over FYI, and –68% over MYI (Fig. 2a). GEOS-Chem also underestimates CALIOP aerosol extinctions over Northern Russia, which could be due to missing sources of aerosols and their precursors from gas flaring in the region (Li et al., 2016; Klimont et al., 2017; Xu et al., 2017).

In the STD simulation, aerosol extinction coefficients during the cold season are dominated by SSA over the high latitude open ocean and by long-range transport of sulfate aerosols over sea ice (Fig. 1f and g). Adding blowing snow emissions of SSA increases the simulated aerosol extinctions by ~10 Mm$^{-1}$ over the central Arctic, bringing the STD+Snow simulation in better agreement with CALIOP observations (Fig. 1c and h), with a model bias of –7% on FYI and –17% on MYI (Fig. 2b). Inclusion of frost flower emissions of SSA in the STD+FF simulation has the largest influence over Canadian Arctic Archipelago where cold temperatures and open leads co-exist (Fig. 1j), but the overall magnitude of the increase cannot explain the CALIOP extinctions. Monthly maps of the comparison between CALIOP and GEOS-Chem simulations are included in the supplement (Fig. S5).

Figure 3 compares the vertical and seasonal distribution of aerosol extinction coefficients in the lower troposphere (0–2 km altitude) over FYI, MYI, and the Canadian Arctic Archipelago, where frost flowers are expected to have their largest influence (Huang and Jaeglé, 2017). Over all three regions, the STD simulation underestimates cold season CALIOP extinctions by factors of 3–6 (Fig. 3d and f). The STD+FF simulation reduces the negative model bias, but the modeled extinctions remain

too low by 20–40% (Fig. 3d–f). Furthermore, the STD+FF simulation does not capture the rapid increase in CALIOP extinctions in October–December (Fig. 3g–i). We find that applying a single scaling factor to the frost flower emissions cannot address the seasonally-varying model discrepancy. In comparison, the STD+Snow simulation displays the best agreement with the CALIOP observations, reproducing both the vertical profile and seasonal cycle of CALIOP extinctions. The STD+Snow simulation, however, underestimates the CALIOP aerosol extinctions in October–December by 30–50% (Fig. 3g and f), and underestimates the surface CALIOP cold season aerosol extinctions by up to 5 Mm$^{-1}$ (Fig. 3d and e). In addition, it predicts a maximum in aerosol extinction during April, while CALIOP observations display their largest concentrations in January–March for FYI, and in March over MYI.

## 3.2 Antarctic

During the Austral cold season (May–October), CALIOP aerosol extinction coefficients decrease with increasing latitudes, ranging from 20–30 Mm$^{-1}$ at 60ºS to 5–10 Mm$^{-1}$ near coastal Antarctica (Fig. 4a). Over Antarctic FYI (Fig. 5a), CALIOP aerosol extinctions display values of 10–14 Mm$^{-1}$ in July–October in the lower troposphere (Fig. 5g), with aerosol extinctions attaining 30 Mm$^{-1}$ near the surface (Fig. 5d). Over MYI sea ice offshore of the Ronne ice-shelf, the CALIOP extinctions are somewhat smaller, reaching 20 Mm$^{-1}$ near the surface (Fig. 5e).

Poleward of 70°S, open ocean SSA dominate aerosol extinction coefficients in the STD simulation (Fig. 4f), accounting for 80% of the extinction, with the remaining 20% due to the combined contributions from sulfate, black carbon and organic aerosols. The STD simulation underestimates CALIOP observations by 5–10 Mm$^{-1}$ (Fig. 5g–i), with a –53% bias over FYI and a –64% bias over MYI (Fig. 2d). The inclusion of frost flowers in the STD+FF simulation leads to a ~2 Mm$^{-1}$ increase in extinction coefficients near the Ross and Ronne Ice-shelves where cold temperatures and open leads persist (Fig. 4i). This increase is insufficient to explain CALIOP observations (Fig. 4e and 5).

The STD+Snow simulation increases aerosol extinction coefficients by 10-20 Mm$^{-1}$ in the Indian Ocean (0–100°) and Pacific Ocean (180–270°) sectors (Fig. 4g), where strong winds persist. We find that the inclusion of blowing snow SSA emissions results in a 43% overestimate of CALIOP extinctions over FYI sea ice (Fig. 2e) and too strong a seasonal increase in extinctions between May and October (Fig. 5g). Over the smaller MYI region, the positive bias of the STD+Snow simulation is +26% (Fig. 2e, 5e and h). Monthly maps comparing CALIOP and GEOS-Chem are included in supplement (Fig. S6).

## 4 Blowing snow simulation with optimized snow salinity

While the inclusion of blowing snow leads to improved agreement with CALIOP, we hypothesize that the remaining discrepancies in the magnitude and seasonal cycle of aerosol extinction coefficients are due to our simplifying assumption of a uniform surface snow salinity over FYI.

As discussed in Section 1, the surface snow salinity is highest over thin FYI with little snow cover early in the cold season, declining in the ensuing months as a result of thickening sea ice and increasing snowpack depth (Barber et al., 1995; Krnavek et al., 2012; Weeks and Lee, 1958; Weeks and Ackley, 1986; Nakawo and Sinha, 1981). As no systematic observations of surface snow salinity are available over sea ice, our approach is to find the monthly salinity of snow on FYI required to minimize the discrepancy between CALIOP extinctions and the GEOS-Chem simulation. The salinity of surface snow on MYI

is the same as in the STD+Snow simulation.

By using a linear regression between the GEOS-Chem simulation and CALIOP monthly extinctions over Arctic sea ice, we derive a surface snow salinity on FYI of 0.9 psu in September, decreasing to 0.09 psu in April (with values of 0.36, 0.26, 0.19, 0.16, 0.16, and 0.14, between October and March). The inferred snow salinities decrease with time and are generally consistent with observations near Alaska reported by Krnavek et al. (2012): 0.8 psu for snow over recently frozen thin FYI and 0.1 psu

over thick FYI. For Antarctic FYI, we infer snow salinities of 0.05 psu in April, 0.02 in May–June and 0.018 in July–September. For the rest of the year the salinity is 0.015 psu. These decreasing trends in salinity between fall and spring are consistent with expectations based on the seasonal evolution of FYI thickness, sea ice surface salinity and deepening snow cover (Worby et al., 1998; Warren et al., 1999; Massom et al., 2001; Kwok and Cunningham, 2015).

We use our empirically-derived monthly snow salinities to conduct an optimized blowing snow simulation (STD+Opt. Snow).

Over Arctic FYI, the model bias changes from –7% (STD+Snow) to +8% (STD+Opt. Snow), and for MYI the model bias of –17% changes to –2% (Fig. 2b and c). Over Antarctic FYI, the model overestimate decreases from +43% (STD+Snow) to +4% (STD+Opt. Snow). Similarly, the model bias over MYI decreases to –10% (Fig. 2e and f). The STD+Opt. Snow simulation displays cold season extinction profiles that are within 5–10% of CALIOP observations over sea ice (Fig. 3d–e and 5d–e). The seasonal cycles are in better agreement with observations, especially in the Arctic for October–December when the

inferred FYI salinities (0.36-0.19 psu compared to 0.1 psu in the STD+Snow simulation) lead to a near doubling of aerosol extinction (Fig. 3g–h). Over Antarctica, the amplitude of the seasonal cycle of aerosol extinction over FYI is reduced, in better agreement with CALIOP observations (Fig. 5g–h).

We also examined whether a single fixed value of salinity over FYI can lead to similar improvements in the agreement with CALIOP. The resulting fixed salinities are 0.11 psu over Arctic FYI and 0.018 psu over Antarctic FYI, leading to good overall

agreement with CALIOP over the Antarctic (NMB of +5% on FYI and –9% on MYI), but no significant improvement seen in the Arctic (NMB of –7% on FYI and –18% on MYI). We found that over the Arctic, a simulation using a single salinity of

0.11 psu (STD+Const. Snow, Fig. S8g–h) yields results similar to the STD+Snow simulation and cannot explain the high extinction values during fall/early winter. Over Antarctic sea ice, the performance of a simulation with 0.018 psu over FYI shows results similar to the STD+Opt. Snow simulation. Thus there is a stronger case for using a seasonally varying snow salinity over Arctic sea ice than over Antarctic sea ice. We speculate that this might be linked to relatively smaller seasonal variation in sea ice thickness and snow depth for Antarctic sea ice compared to the Arctic. In their snow climatology, Warren et al. (1999) report that the mean snow depth at an Arctic sea ice site increased from 8.7 cm in October to 28.9 cm in March. Satellite-based observations of Arctic FYI thickness show an increase from 0.95 m in October to 2.15 m in May (Kwok and Cunningham, 2015). In contrast, over Antarctic sea ice the mean sea ice thickness and snow depth remained fairly constant during fall–winter (April: 0.48 m for ice thickness and 0.11 m for snow depth; August: 0.52 m for ice thickness and 0.11 m for snow depth) as described in Worby et al. (1998).

Both STD+Snow and STD+Opt. Snow simulations underestimate CALIOP aerosol extinctions over the Canadian Arctic Archipelago (Fig. 3f). Combining the STD+Opt. Snow and frost flower emissions could help improve the agreement in that region, but it would also lead to substantial overestimates over FYI and MYI for the rest of the Arctic. One possibility is that snow-covered frost flowers in the Canadian Archipelago increase the local surface snow salinity (Domine et al., 2004). The recent study of Hara et al. (2017) in northwestern Greenland proposed that snowfall buries frost flowers and the associated slush layer on new FYI. The resulting brine migrates vertically enriching the surface snow layer, which can be mobilized under strong winds. We estimate that increasing the salinity of snow over the Canadian Archipelago to a value of 3 psu would help reconcile the optimized blowing snow simulation with CALIOP observations. Direct measurements of snow salinity in this region would help confirm this estimate. Similarly, we find that increasing the snow salinity near the Ross ice-shelf region, where frost flowers are expected to occur, would improve the agreement with CALIOP observations.

Figure 6 evaluates the performance of the STD+Opt. Snow simulation against independent observations of SSA mass concentrations at Barrow, Alaska (71.3ºN, 156.6ºW); Alert, Nunavut, Canada (82.5ºN, 62.5ºW); Zeppelin, Svalbard (78.9ºN, 11.9ºE); Neumayer, Antarctica (70.7°S, 8.3°W) and Dumont d'Urville, Antarctica (66.7ºS, 140ºE). Descriptions of the in situ observations are provided in Huang and Jaeglé (2017). At Barrow, the optimized simulation improves the agreement with observed SSA mass concentrations in November–May. In particular, the enhanced salinity in October–November brings the model closer to the observations. At Zeppelin and Dumont d'Urville, the model bias in the STD+Opt. Snow simulation is reduced relative to the STD+Snow simulation. However, the model bias worsens at Neumayer (STD+Snow: –25%, STD+Opt. Snow: –48%), and the model bias remains large at Alert (STD+Snow: –50%, STD+Opt. Snow: –32%). As Neumayer and Alert are close to the frost flower producing regions, compared to other polar sites, this underestimate may be related to an underestimate in the snow salinity in those regions.

It is also possible that the discrepancies between observed and modeled aerosol extinction coefficients are due to other factors in the blowing snow parameterization as implemented in GEOS-Chem. For example, our simulation does not include the

negative feedback of water vapor sublimation (Mann et al., 2000): as blowing snow particles sublime in unsaturated air, they cause an increase in water vapor and thus cooling of the surrounding air. Both effects lead to an increase in RH near saturation, reducing the sublimation rate. Another underlying assumption is that 5 SSA are produced for each snowflake that sublimes (N=5). We conducted a sensitivity simulation assuming one SSA per snowflake, shown as STD+Snow (N=1) in the supplement (Fig. S7 and S8). This change does not affect the total emissions of blowing snow SSA, but decreases the fraction of SSA in the accumulation mode (see Huang and Jaeglé, 2017). As the extinction efficiency of accumulation mode SSA is larger than that of coarse mode SSA, the assumption of N=1 leads to a 30–50% decrease in modeled extinctions relative to the STD+Snow (N=5) simulation. Overall, this results in improved agreement with CALIOP observations over Antarctic sea ice, but the CALIOP aerosol extinctions are underestimated over the Arctic. Increasing the surface snow salinity over Arctic FYI can address the model discrepancy in aerosol extinction coefficients, but will lead to a factor of 1.5–2 overestimate in SSA mass concentrations.

## 5 Case study of a blowing snow event over the Arctic

Figure 7 shows a case study of a blowing snow SSA event, which occurred on November 6, 2008 over the Arctic. The STD+Snow and STD+Opt. Snow simulations display enhanced extinction coefficients (40–80 Mm$^{-1}$) over the Barents Sea along the 60ºE longitude line extending from the North Pole to 70ºN (Fig. 7b and c). This feature is due to blowing snow SSA and it is not seen in the STD simulation (Fig. 7a) or the frost flower simulation (not shown).

The CALIPSO 00:58–01:12 UTC overpass on November 6, 2008 transected this feature, with CALIOP aerosol extinction coefficients of 50–150 Mm$^{-1}$ between points A (78°N; 52.5°E) and B (82° N; 110° E) labelled in Fig. 7a–c. The cross section along the CALIOP overpass shows that the large extinctions are confined between the surface and 1–2 km altitude (Fig. 7e). This overpass region is mostly covered by FYI (Fig. 7d). We sampled GEOS-Chem along the same cross section, finding a very good correspondence in the spatial extent of the feature observed by CALIOP and simulated by the STD+Snow and STD+Opt. Snow simulations (Fig. 7g–i). The optimized blowing snow model predicts higher aerosol extinctions due to larger surface snow salinity on FYI in November, and is in better agreement with the 0–2 km CALIOP mean aerosol extinction coefficients (Fig. 7i).

We use the FLEXPART particle dispersion model (Stohl et al., 1998, 2005; Seibert and Frank, 2004) with meteorological data from ERA-Interim with a horizontal resolution of 0.5° to track the origin of this feature. We release 100,000 particles between points A and B at 0.01–2km and track them back in time over 2 days. The air along the AB transect in Fig. 7 originates from the 120–140°E sector at 60°–90°N near the surface (0–100 m), on November 4, 2008 (Fig. 8a). This region is covered by both MYI and FYI (Fig. 7d). The STD+Snow model predicts enhanced blowing snow emissions in this region on November 4, 2008 (Fig. 8b). The CD transect in Fig. 8b shows the CALIOP overpass at 01:11–01:24 UTC November 4, 2008. This CALIOP

transect displays elevated 532 nm attenuated perpendicular backscatter and depolarization ratio below 200 m (Fig. 8c and d), which are co-located with strong surface winds (>5 m s$^{-1}$). The pattern of elevated backscatter, surface winds and depolarization ratio satisfies the requirements of the CALIOP blowing snow detection algorithm described in Palm et al. (2011, 2017), who defined blowing snow events as layers with high color ratio (>1), high depolarization ratio (>0.25), strong surface winds (>4 m s$^{-1}$), and enhanced backscatter signals below 300 m (ranging between $2.5 \times 10^{-2}$ km$^{-1}$ sr$^{-1}$ and 0.2 km$^{-1}$ sr$^{-1}$). Therefore, the FLEXPART-predicted source region and CALIOP blowing snow feature are co-located with enhanced blowing snow emissions in the GEOS-Chem simulation (Fig. 8b). This case study thus shows that CALIOP can detect not only the blowing snow event (Palm et al., 2011) but also the resulting SSA produced after sublimation.

## 6 Discussion and conclusions

In this work, we used the GEOS-Chem chemical transport model to assess the ability of the CALIOP Lidar onboard the CALIPSO satellite to provide constraints on sea ice sources of SSA. We find that mean CALIOP aerosol extinction coefficients below 2 km altitude reach 10–15 Mm$^{-1}$ over sea ice covered regions during the 6-month polar cold season. The enhanced extinctions are located below 2 km, with the largest values (20–35 Mm$^{-1}$) occurring near the surface. We find that a standard GEOS-Chem simulation without sea ice sources of SSA underestimates CALIOP extinctions by 50–70% over Arctic and Antarctic sea ice. A simulation with frost flower SSA emissions is unable to explain the spatial and temporal distribution of CALIOP aerosol extinctions. Adding a blowing snow SSA source results in improved agreement over the Arctic (NMB = – 7% for first year sea ice (FYI) and NMB=+17% over multi-year sea ice (MYI) ), but yields a 43% overestimate of CALIOP extinctions over Antarctic sea ice. Additionally, the simulation including blowing snow SSA tends to underestimate CALIOP observations during fall–early winter (October–December over the Arctic and April over the Antarctic).

We hypothesize that our assumption of constant surface snow salinity on FYI (0.1 psu over the Arctic and 0.03 psu over the Antarctic) in the blowing snow simulation could explain the remaining discrepancies between observed and modeled extinctions. Given the paucity of snow salinity observations, we infer the monthly surface snow salinity on FYI required to minimize the discrepancy between CALIOP extinctions and the GEOS-Chem simulation. The resulting snow salinities decrease progressively between the beginning and end of the cold season (from 0.9 to 0.09 psu for Arctic FYI; 0.05 to 0.018 psu over Antarctic FYI). This decrease is consistent with the seasonally increasing sea ice thickness and accumulating snow depth. The optimized blowing snow model using the monthly-varying snow salinities shows improved agreement with CALIOP observations and in situ observations of SSA mass concentrations at five surface sites. However, the optimized blowing snow model tends to underestimate the aerosol extinctions over the Canadian Arctic Archipelago and off the Ross Ice-shelf. Both regions are predicted to favor frost flower growth, which could locally increase the salinity of snow when frost flowers are buried under snow. We find that increasing the Canadian Arctic Archipelago FYI snow salinity to 3 psu would help reconcile our simulation with CALIOP and in situ observations. Our work, however, cannot rule out other alternative

factors contributing to the discrepancy between modeled and observed aerosol extinctions, such as the impact of the negative feedback of water vapor sublimation and our assumption about the number of particle produced per snowflake. Systematic observations of surface snow salinity over multiple sea ice locations and times would help further constrain snow salinities in the Arctic and Antarctic. Furthermore, more extensive observations of sea salt aerosol size distributions during blowing snow

events could help further refine and constraint these assumptions.

We conduct a case study of a blowing snow SSA event over the Arctic which was detected by CALIOP on November 6, 2008 and predicted by our blowing snow simulation. Using FLEXPART, we find that the observed aerosol extinction layer originated 2 days earlier over sea ice below 100m altitude. We demonstrate that CALIOP detects this blowing snow event with enhanced extinction below 200 m and large depolarization ratio, co-located with surface high winds.

Our work suggests that blowing snow emissions are the dominant source of SSA over sea ice covered regions during cold months. As SSA can act as a source of halogens, inclusion of blowing snow in chemical transport models is important to understand springtime bromine explosions and the resulting ozone and mercury depletion events (Schroeder et al., 1998; Simpson et al., 2007; Steffen et al., 2008; Gilman et al., 2010; Yang et al., 2010). Furthermore, these sea-ice sources of SSA can act as ice nuclei for cloud formation, and may increase the downward longwave radiative forcing (Xu et al., 2013). Arctic

sea ice has been rapidly changing over the past 30 years, with decreasing sea ice extent and thickness (e.g. Kwok and Rothrock, 2009), a shift towards less MYI and more FYI (Fowler et al., 2004; Maslanik et al., 2007), and a thinning of snow depth in spring (Renner et al., 2014; Blanchard-Wrigglesworth et al., 2015). All these factors are likely to have resulted in an increase in snow salinity on sea ice, and hence increasing SSA emissions from blowing snow. In the Southern Hemisphere, sea ice extent is FYI-dominant, and it annual/seasonal trend is more complex, varying in space. Sea ice extent has been increasing

over the Ross Sea, but decreasing over the Amundsen-Bellingshausen Sea over the past decades (Turner et al., 2009; Parkinson and Cavalieri, 2012; Stammerjohn et al., 2012). Consequently, this may have resulted in a shift in the spatial pattern of blowing snow SSA emissions, with increased influence over the Ross Sea and reduced influence over the Amundsen-Bellingshausen Sea.

**Acknowledgements.**

This work was supported by funding from the NASA Atmospheric Composition Modeling and Analysis Program under award NNX15AE32G. The CALIOP data was obtained from the NASA Langley Research Center Atmospheric Science Data Center.

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

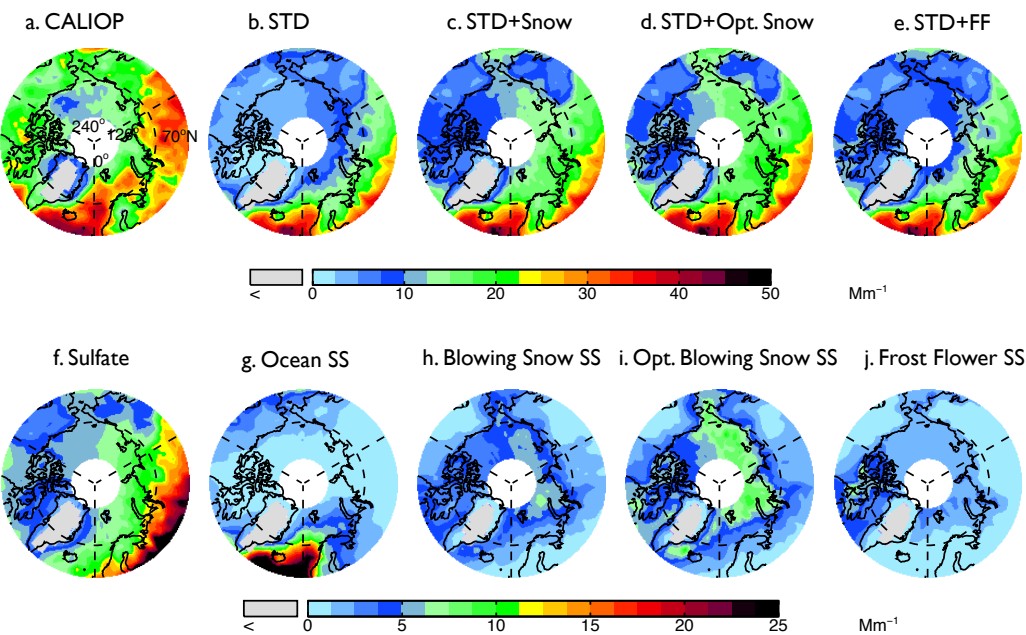

**Figure 1: Spatial distribution of mean aerosol extinction coefficients (0-2 km) during the 2007-2009 Arctic cold season (November-April) observed by (a) CALIOP and calculated with the GEOS-Chem model in (b) a Standard simulation (STD), (c) a simulation including blowing snow SSA emissions (STD+Snow), (d) an optimized blowing snow simulation (STD+Opt. Snow), and (e) a simulation including frost flower SSA emissions (STD+FF). The simulated extinctions are sampled at the time and location of the CALIOP overpasses, and the CALIOP sensitivity threshold is applied. The bottom panels show the extinction coefficients of individual aerosol components in the GEOS-Chem simulations: (f) sulfate aerosol, (g) open ocean SSA, (h) blowing snow SSA, (i) Opt. blowing snow SSA and (j) frost flower SSA. Note the different colorbar scales for the top and bottom rows.**

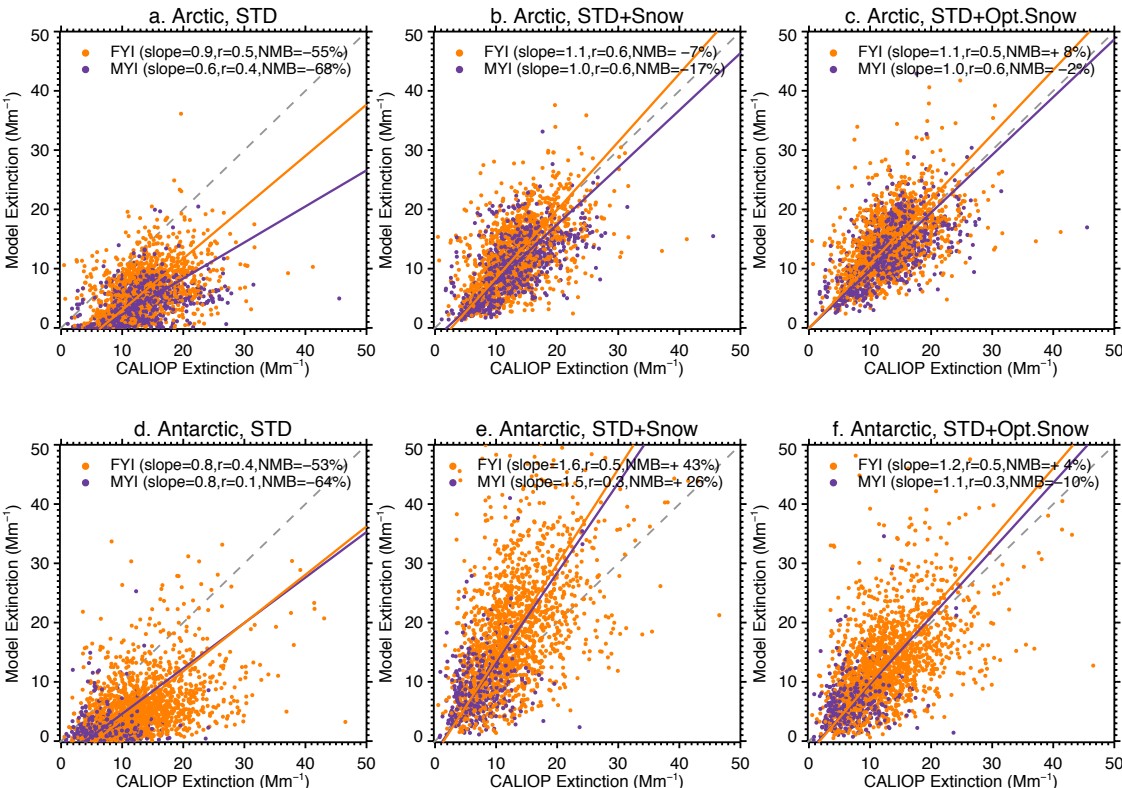

**Figure 2. Scatter plot of CALIOP and GEOS-Chem aerosol extinction coefficients over FYI (orange circles) and MYI (purple circles) for the Arctic cold season (November−April, top panels) and Antarctic cold season (May–October, bottom panels). Each symbol represents the monthly aerosol extinction coefficients for individual grid boxes (2°x5°, 0–2 km) over sea ice. The dashed gray line is the 1:1 line. The purple and orange lines are the linear fit for the points over MYI and FYI, respectively. The slope of the regression line, correlation coefficient (r) and normalized mean bias, NMB $= 100\times(\overline{Model}/\overline{Obs} - 1)$, are given in the insert of each figure.**

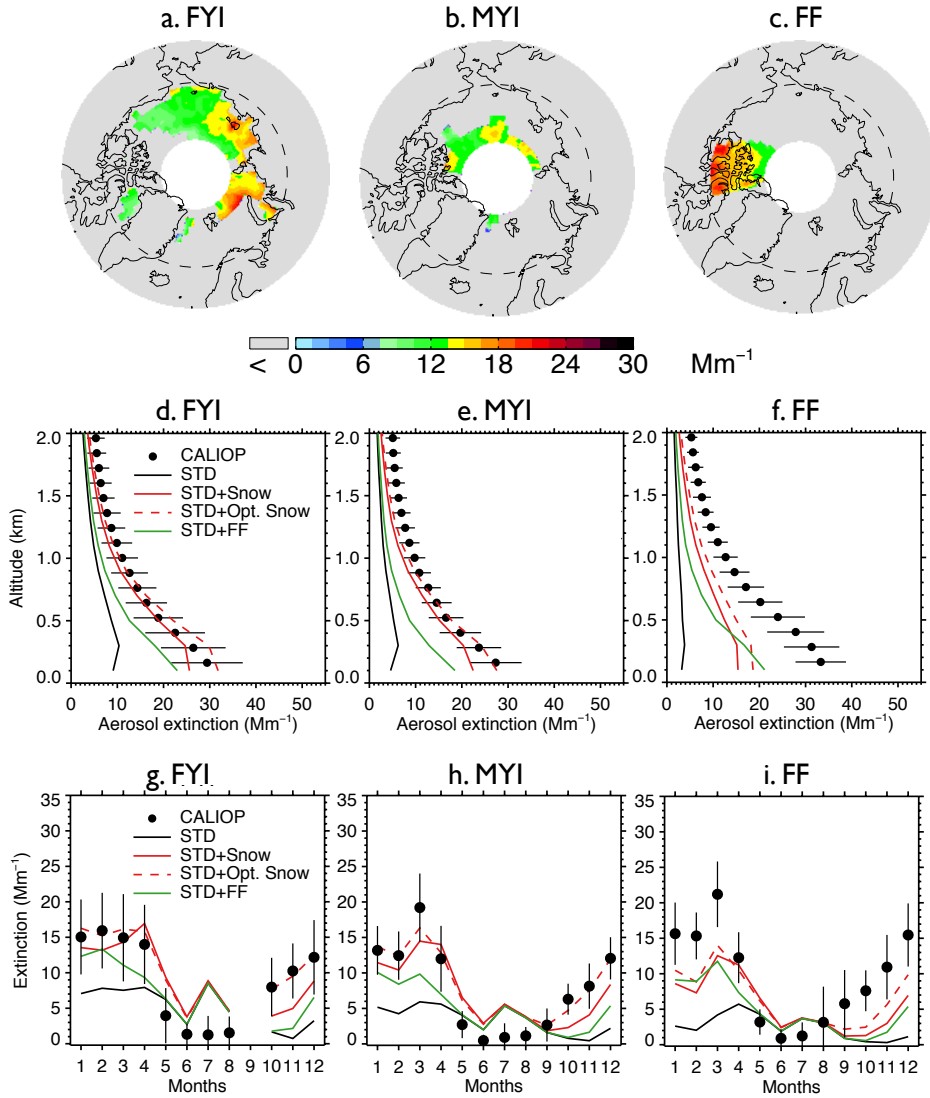

**Figure 3: Top row: 2007–2009 CALIOP mean aerosol extinction coefficients (0–2 km) in the Arctic cold season (November–April) over (a) first-year sea ice (FYI), (b) multi-year sea ice (MYI) and (c) the Canadian Arctic Archipelago (CAA). Middle row: Vertical profiles of Arctic cold season mean aerosol extinction coefficients over (d) FYI, (e) MYI and (f) CAA for CALIOP (black dots with horizontal lines indicating standard deviations) and GEOS-Chem model simulations (STD: black lines, STD+Snow: red solid lines, STD+Opt. Snow: red dashed lines; STD+FF: green lines). Bottom row: Seasonal cycle of 0–2 km monthly aerosol extinction coefficients averaged over (g) FYI, (h) MYI and (f) CAA. CALIOP observations are shown as black circles and vertical lines indicate the interannual standard deviation. The four GEOS-Chem model simulations are also shown (STD: black lines, STD+Snow: red solid lines, STD+Opt. Snow: red dashed lines, STD+FF: green lines).**

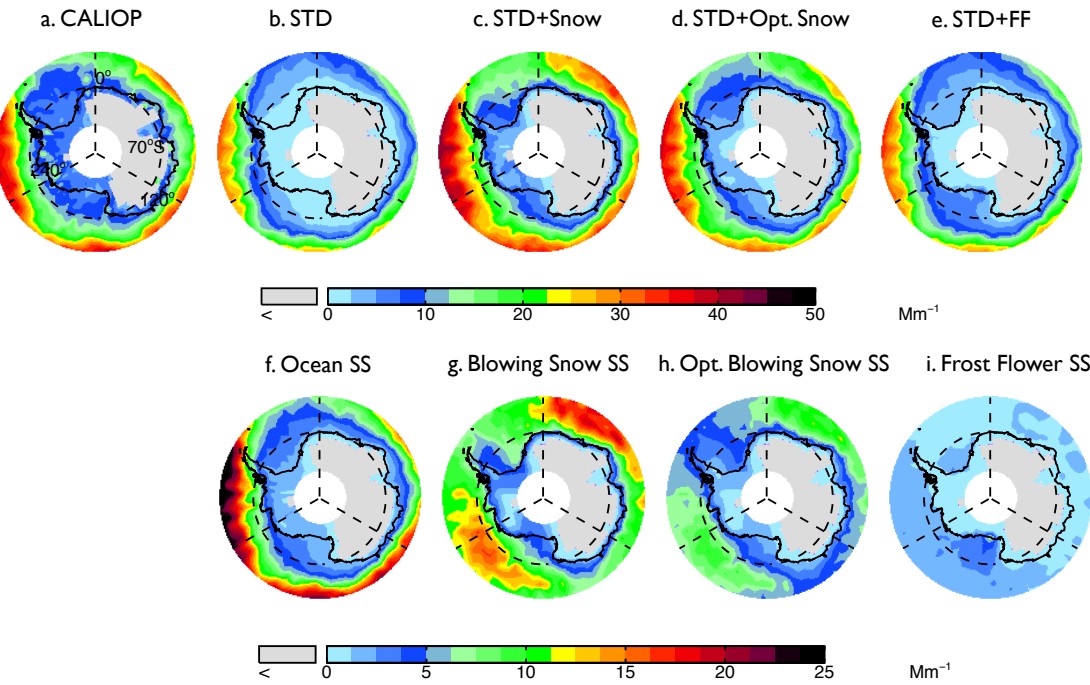

**Figure 4. Spatial distributions of mean aerosol extinction coefficients (0–2 km) during the 2007-2009 Antarctic cold season (May–October) observed by (a) CALIOP and calculated with the GEOS-Chem (b. STD, c. STD+Snow, d. STD+Opt. Snow, e. STD+FF). The bottom panels show extinctions of individual aerosol components in the GEOS-Chem simulations: (f) open ocean SSA, (g) blowing snow SSA, (h) Opt. blowing snow SSA and (i) frost flower SSA. Note the different colorbar scales for the top and bottom rows.**

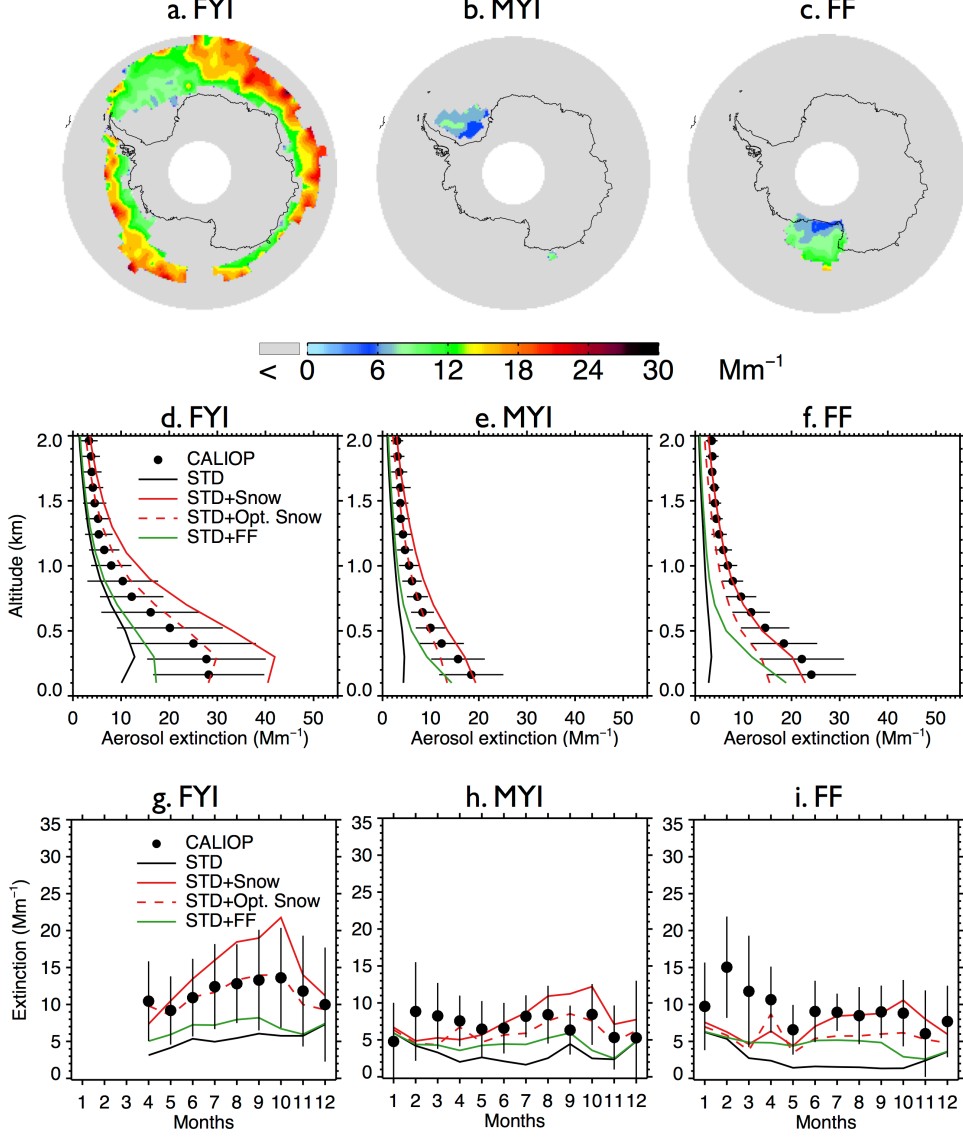

**Figure 5. Same as Figure 3, but for Antarctic aerosol extinction coefficients during Austral winter (May–October) over (a) FYI (excluding offshore Ross Ice-shelf), (b) MYI and (c) offshore Ross Ice-shelf. As shown in (g), the monthly average aerosol extinction coefficients are not available over FYI during Antarctic summer (January–March) due to the limited FYI extent.**

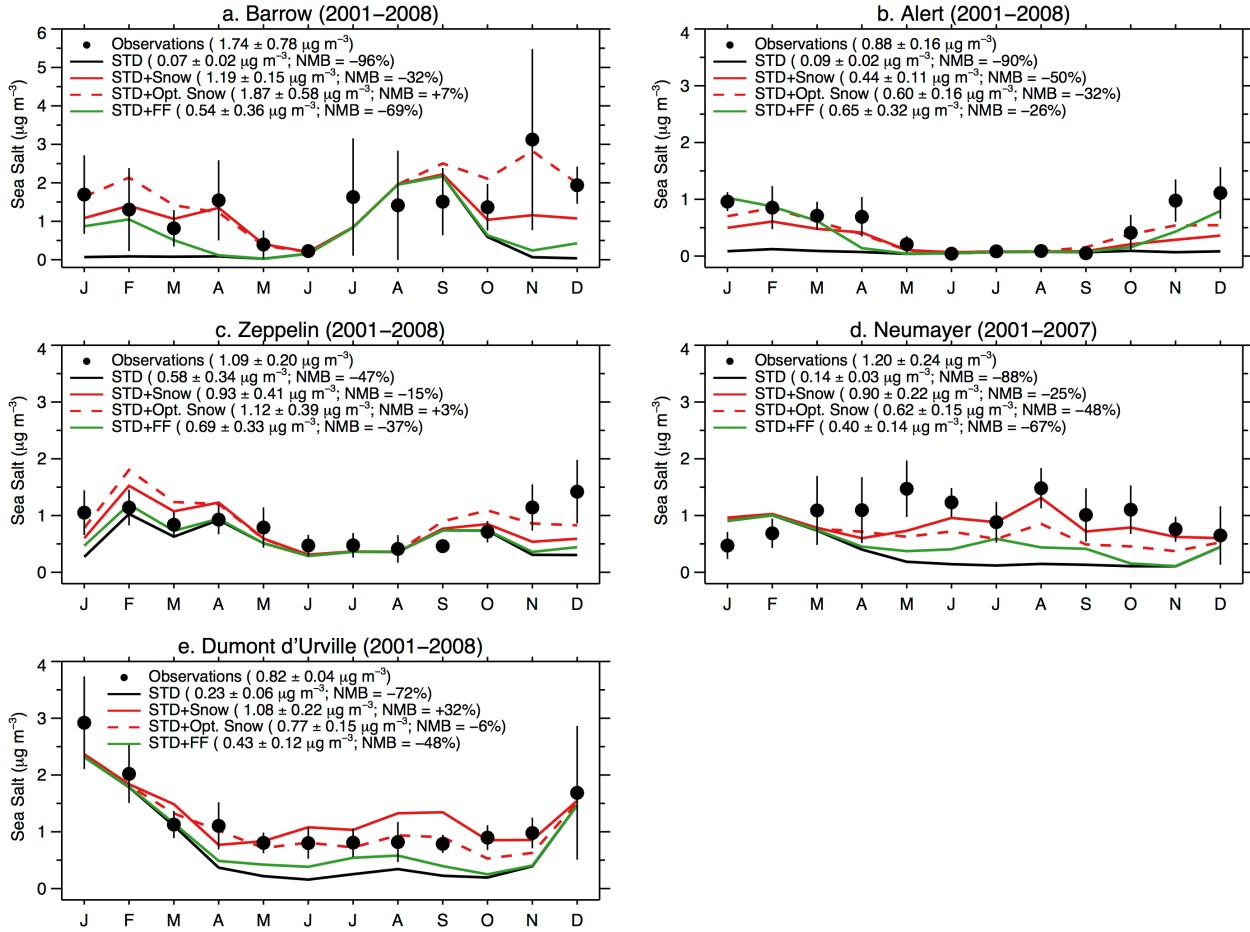

**Figure 6. Monthly mean SSA mass concentrations at Arctic sites (a. Barrow, b. Alert, c. Zeppelin) and Antarctic sites (d. Dumont d'Urville, e. Neumayer). All observations and model results are for 2001–2008 except at Neumayer (2001–2007). The observed mean concentrations are indicated with filled black circles, and the lines are for the GEOS-Chem simulations (STD: black line, STD+Snow: red line, STD+Opt, Snow: red dashed line; STD-FF: green line). The black vertical lines are the standard deviations of monthly means for the observation years. For each individual panel, the insert lists mean concentrations and standard deviations, as well as the NMB for the cold season only (Arctic: November–April; Antarctic: May–October).**

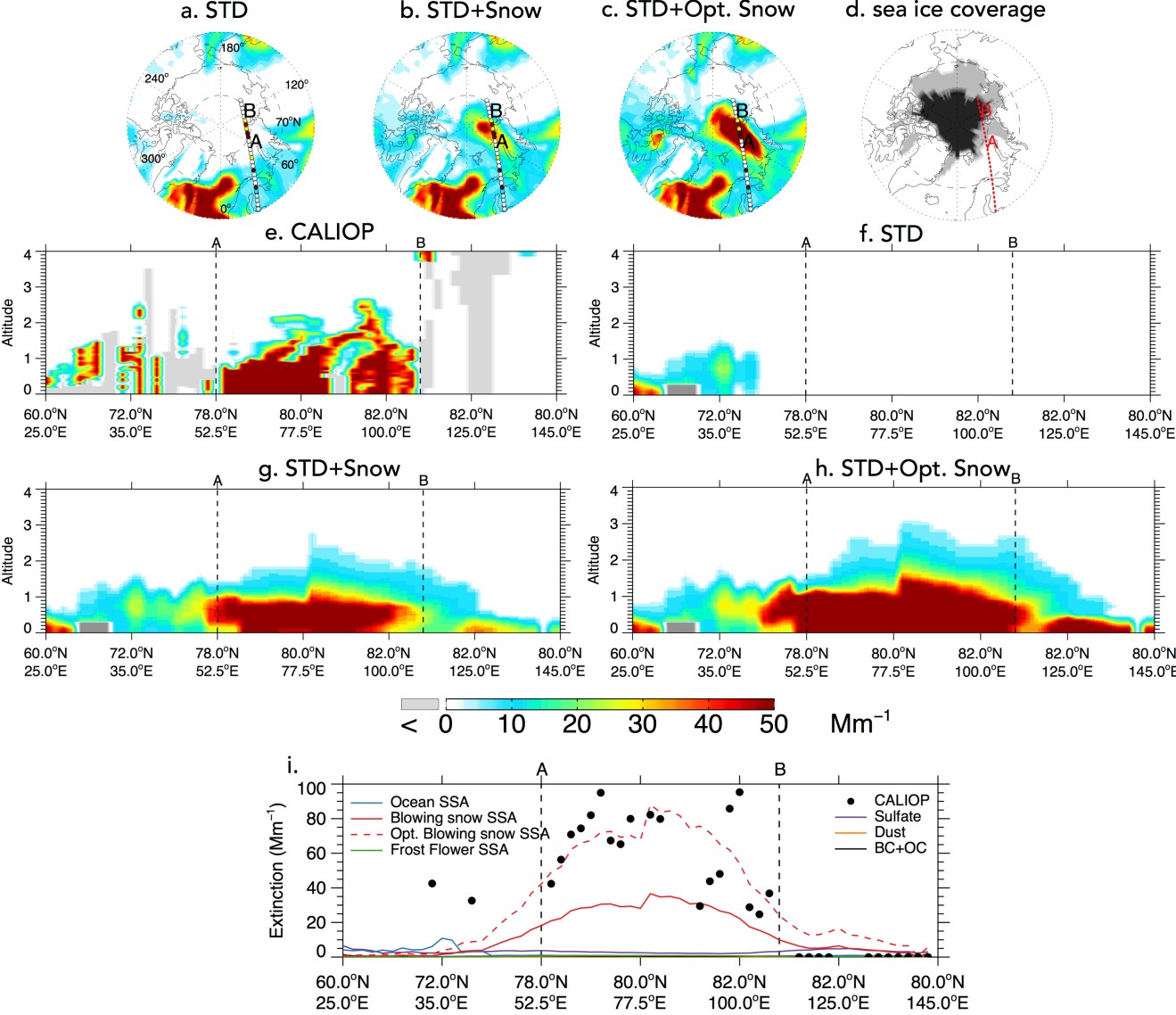

**Figure 7. November 6, 2008 case study of a blowing snow SSA feature over the Arctic. Top panels: Spatial distribution of mean aerosol extinctions below 2 km altitude for the (a) STD, (b) STD+Snow, and (c) STD+Opt. Snow GEOS-Chem simulations. The CALIOP nighttime overpass at 00:58–01:12 UTC is displayed on the top panels, with filled circles color-coded according to observed mean extinction coefficients below 2 km. The overpass intercepts the SSA blowing snow feature between points A and B. Panel (d) shows the MERRA sea ice coverage on November 6, 2008, with light gray shading indicating FYI and black shading for MYI. Panels (e–h) show the observed and simulated vertical cross-sections of aerosol extinction coefficients along the CALIOP overpass. The light gray shading in (e) indicates that no valid data is available for CALIOP. The dark gray shading in (f–h) shows the local topography in the model. Panel (i) shows the 0–2 km CALIOP mean aerosol extinctions and the contributions of different aerosol types in the GEOS-Chem simulations: sulfate aerosol, dust, black carbon and organic carbon (BC+OC), open ocean SSA, blowing snow SSA, Optimized blowing snow SSA, Frost Flower SSA.**

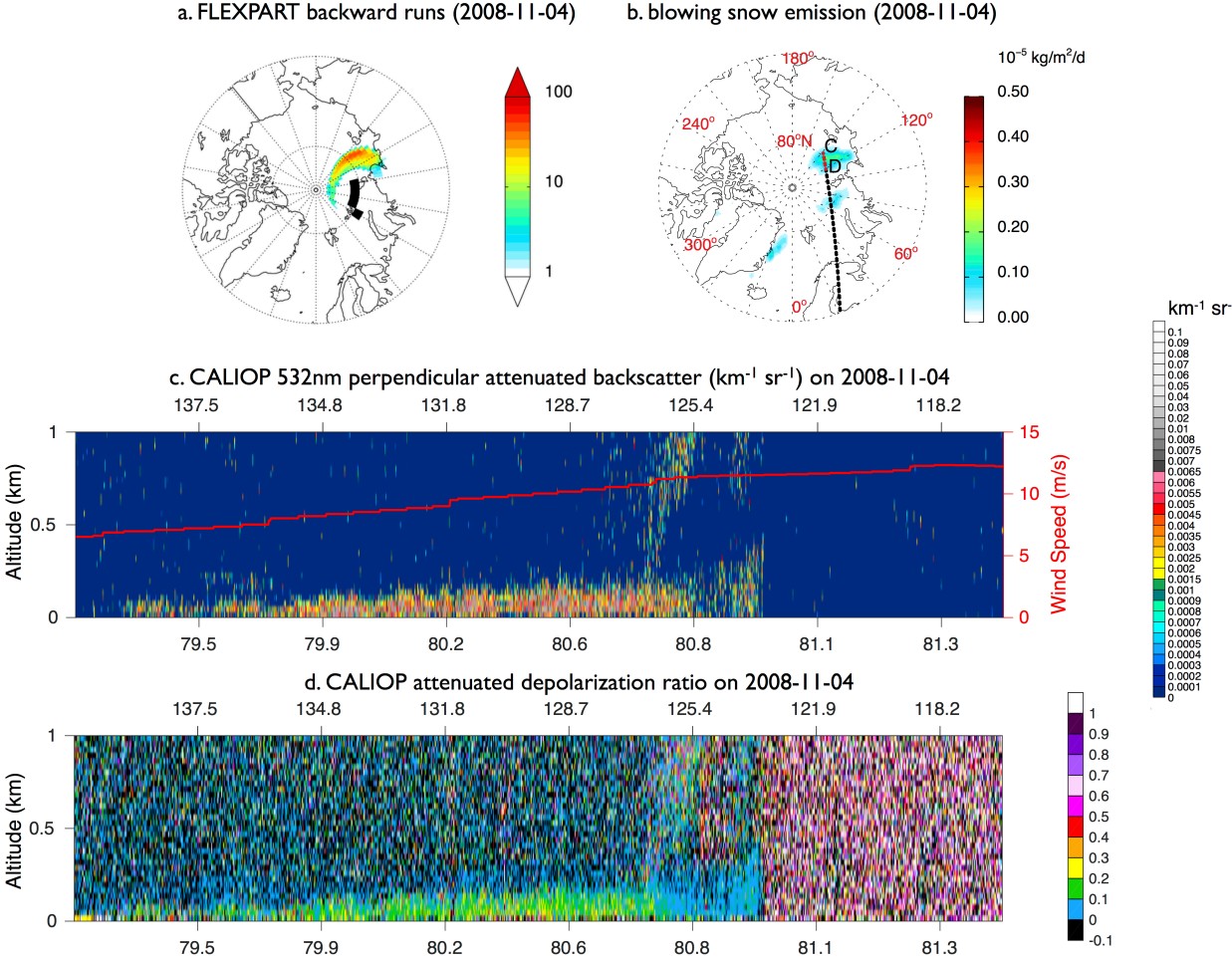

**Figure 8. (a) The November 4, 2008 FLEXPART footprint below 100m (in seconds) for particles initialized at 0-2 km over the black squares on November 6, 2008, near the blowing snow feature observed by CALIOP (Figure 7). (b) November 4, 2008 blowing snow SSA emissions from the STD+Snow simulation, and the CALIOP overpass at 01:11–01:24 UTC on that day. The bottom two panels display CALIOP cross-sections between points C and D for (c) the 532nm perpendicular attenuated backscatter (km⁻¹ sr⁻¹) and (d) attenuated depolarization ratio. The surface wind speed (m s⁻¹) is shown with a red line in panel c.**