# Peer review of "Using CALIOP to constrain blowing snow emissions of sea salt aerosols over Arctic and Antarctic sea ice"

_Atmospheric Chemistry and Physics, 2018_

## Referee Comment (RC1) · Anonymous Referee #2 · 5 Jun 2018

This is a well written manuscript describing modeling of Arctic aerosol and comparison of these models to observations from CALIOP satellite lidar observations. The model, GEOS-Chem, uses various parameterizations of aerosol production mechanisms, and addition of a blowing snow mechanism brings the model closer to observations. The blowing snow model is further refined by varying the surface snow salinity to improve agreement with observations. An example of an event of blowing snow is shown.

Overall, I feel that this is a well written manuscript, but that the identification of model modifications with specific physical processes sometimes goes further than is justified and/or alternative hypotheses have not been explored fully. The CALIOP data indicate

[Figure]

**[ACPD](ACPD)**

Interactive
comment

that there is larger extinction present near the surface than the model would indicate, so a wind speed and snow salinity dependent blowing snow model is added, increasing the modeled aerosol extinction, which brings it closer to observations. However, one needs to consider how definitive the identification of these model variables is with physical processes. Specific questions in this regard are:

1) After adding "blowing snow", the model is tuned to reduce surface snow salinity in MYI areas as compared to FYI areas and over the wintertime season. How robust is the necessity to tune down the salinity? For example, Figure 3 shows distributions of extinction in FYI, MYI, and CAA areas. Visually, I can barely see any difference between the CALIOP observations in panels g, h, and i. Values are about 15 Mmˆ-1 from Jan-Apr, low in summer, and increase back to 15 Mmˆ-1 towards the end of the year. Is there any statistical difference between these monthly observational distributions? Given the lack of difference between these locations, it seems like the need to optimize the model is weak. Specific monthly values are listed, but it doesn't seem like there is enough information to actually map out this amount of information. For example, could a different single fixed value of salinity be used to optimize the model similarly? It is not unreasonable that surface snow salinity would decrease as you add new snow (which is of low salinity), but the question is how strong the modeling evidence for this decrease is. Please show that the trend from the "optimization" is a real effect larger than statistical errors.

2) Open water areas can produce aerosol directly (by wind blowing over the exposed sea water) or via re-freezing, which might produce frost flowers and/or simply provide a non-snow-covered highly saline surface that snow could blow onto/across. The manuscript does not do justice to hypotheses other than frost flowers. It should leave open the possibility that open water or thin snow cover on ice could be responsible. For instance, the citation below indicates that open water is a source of sea salt aerosol.

May, N. W., P. K. Quinn, S. M. McNamara, and K. A. Pratt (2016), Multiyear study of the dependence of sea salt aerosol on wind speed and sea ice conditions in the coastal

Arctic, J. Geophys. Res. Atmos., 121, 9208–9219, doi: 10.1002/2016JD025273.

Another aspect that may affect the ability to model either open water areas of frost flowers is the low spatial resolution (2 x 2.5 degree) of sea ice in the model and also the use a weekly product (Page 5, line 30) for sea ice concentration. This low time resolution and linear interpolation could affect the ability of the model to represent the small spatial scale (few km) and temporally transient sea ice lead features.

3) The Canadian Archipelago is a region where there is a great deal of land near sea ice. The land can affect the ability of passive microwave satellites to detect sea ice concentrations (called land contamination), and thus could affect the ability to predict frost flower presence. Also, surface winds in the presence significant topography might not be modeled well at these course spatial resolutions. Therefore, I think that there may be a number of factors in this region and caution against overinterpretation. For example, page 9, line 7 indicates a surface snow salinity of 3 psu (nearly 10% of that of sea water) could reconcile differences. Also, it is stated that Alert is near frost-flower producing regions. I think of Alert being in a MYI area, largely surrounded by older sea ice that builds over years. Please cite sources to indicate evidence for Alert (and Neumayer) being in frost-flower producing area.

Minor comments:

Page 2, line 20. This sentence is somewhat confusing with respect to what surface is being discussed. Is the top of the newly forming first year ice's salinity being discussed? If so, please clarify that this is the ice surface rather than snow.

Page 3, line 3. There is no discussion of open water as a sea salt source.

Page 3, line 27. The wording of "aerosol extinctions and the layers beneath" maybe could be improved.

Page 4, line 20. I think it should be "...with a 1-year..."

Page 6, line 26. The wording of "reducing the bias" maybe could be improved (the bias

became larger, not smaller, but closer in magnitude to zero).

Overall, I feel that this manuscript argues well for the need to add a wintertime sea salt aerosol source to the Arctic and this source seems to be effectively modeled by a blowing snow model, but that some further refinements of this model may not be appropriately linked to physical processes (e.g. surface snow salinity changes and frost flowers). Those aspects of the manuscript should be further defended by statistical methods or should be written in a more cautious manner, including alternate hypotheses that seem consistent with the data.

---

## Referee Comment (RC2) · Anonymous Referee #1 · 18 Jun 2018

This paper describes atmospheric cycles of sea-salt aerosols in polar regions using model and remote sensing measurement (CALIOP). Authors applied and improve the model, GEOS-chem., to simulate spatial distribution and origins of sea-salt aerosols on basis of various parameters such as salinity of surface snow. They derived an interesting conclusion that sea-salt aerosols in the winter were involved in blowing snow rather than frost flowers on sea-ice. On the whole, the topic of the manuscript is relevant and suitable for the scope of the "Atmospheric Chemistry and Physics. The topics and results deserve to be made available to the scientific community and to be exploited in terms of atmospheric aerosols and ice core community in polar regions. Therefore, this study adds very useful information to our knowledge on the sea-salt cycles in-

volved in blowing snow in polar regions. From this reason, I support publication of this work in ACP. However, the current version contains obvious weaknesses, therefore I recommend a major revision. Details are shown as follows.

1. Size distributions of sea-salt aerosols In the GEOS-Chem. Model, spatial distributions of the concentrations of sea-salt aerosols were calculated on assumption of dry deposition velocity and emission from some origins (e.g., open water, frost flowers, and snow). Sea-salt aerosols were distributed from ultrafine to coarse modes in the polar regions during winter – spring (e.g., Hara et al., ACP, 2011).

Hara, K., et al.: Seasonal features of ultrafine particle volatility in the coastal Antarctic troposphere, Atmospheric Chemistry and Physics, doi:10.5194/acp-11-9803-2011, 2011.

What is procedure to calculate and treat size distributions and concentrations of sea-salt aerosols? What are the initial size distributions of particles immediately after emission from sea-ice and ocean? I think that these parameters are probably as same as those in your previous work (Huang and Jaegle, ACP, 2017). If so, add short explanation about processing of aerosol size distribution in the model for readers. If not, details should be mentioned.

2. Dry deposition velocity In this study, aerosol dry deposition velocity was fixed to 0.03 cm s-1, corresponding to that of particles with size of ca. 2um in diameter. As shown by Rhodes et al. (2017) and Hara et al. (2017), sea-salt aerosols and ice particles containing sea-salts were released from snow and frost flowers on sea-ice. Then, size of sea-salt particles and ice particles containing sea-salts can be changed through sublimation and efficient dry deposition of larger sea-salt particles in the atmosphere. In general, the coarser aerosols have larger dry deposition velocity (shorter residence time). Therefore, processing of initial size distribution and modification of size distribution involved simultaneously with dry deposition and sublimation is the most important to simulate the concentrations and spatial distribution of sea-salt aerosols. Because

aerosol dry deposition velocity has size-dependence, the fixed and assumed aerosol dry deposition velocity can result in mis-estimation. I understand that it is difficult to input all parameters in model calculation. However, sensitivity of dry deposition on the sea-salt concentrations should be checked. Ideally, size dependence of dry deposition velocity is included in the model (I do not require it this time, but I hope it for progress in the future).

Rhodes, R., Yang, X., Wolff, E., McConnell, J. and Frey, M.: Sea ice as a source of sea salt aerosol to Greenland ice cores: a model-based study, Atmospheric Chemistry and Physics, 17(15), 9417–9433, doi:10.5194/acp-17-9417-2017, 2017.

3. Potential frost flower (PFF) coverage PFF were estimated using air temperature and thickness of sea-ice in the study. Actually, frost flower can be formed on new and young sea-ice. In this study, threshold of newly formed sea-ice thickness is 10 cm. In my experience, this value is small, because frost flower can be appeared on sea-ice even with thickness of ca. 30cm. If the threshold was smaller, the model results can be underestimated. What is the impact of sea-ice thickness in the model?

4. Spatial distribution (Figs. 1 and 4) Spatial distribution of aerosol extinction coefficients and model results during cold seasons were depicted in Figs. 1 and 4. These plots provide us very interesting information to understand atmospheric sea-salt cycles in Arctic and Antarctica. However, these periods correspond to develop sea-ice extent, so that these distributions included also seasonal feature of sea-ice extent, which is associated with origins of sea-salt aerosols. Seasonal and spatial variations of source strength and origins of sea-salt aerosols should be taken into account. To exclude influences of the seasonal features, I suggest that the selected months are shown, for example month with maximum of sea-ice extent (March in Arctic and September in Antarctica).

---

## Short Comment (SC1) · 29 Jun 2018

General comments: This manuscript reports a GEOS-chem model study of sea-ice soured SSA (from both blowing snow and frost flowers) and their impacts on polar aerosol extinction. Numerous model results via changing various parameters are performed and compared to remote sensing (CALIPSO) data. Some results are quite interesting, adding novel information to our knowledge regarding polar SSA production. Authors even derive an 'optimized' seasonal trend of snow salinity. Due to the lack of year around blowing snow and snowpack salinity measurements on polar sea ice surface, we almost know nothing about seasonal variation regarding snow salinity.

For this reason, I will treat their modelling-based seasonal snow salinity as a weakness. Instead, I think it highlight an issue which is largely unknown to our knowledge. As we know snow salinity is one of the critical factor that could determine both salt mass loading and their airborne budget (via affecting size spectrum and then lifetime). Therefore, it is a quite important to investigate this parameter in a modelling study, though it needs justification as reviewers pointed out. In general, this is well written manuscript with some interesting results presented. It fits well the scope of the 'Atmospheric Chemistry and Physics' and will benefit relevant communities in sea ice, ice core and boundary layer chemistry. Thus, I support publication of this work in ACP after a revision (see below my specific comments). Specific comments: The STD+Snow model run overestimates satellite extinction coefficients. Authors attribute this overestimation to 'higher' snow salinity applied in their model. However, I notice that the salinity levels of 0.1 psu for the Arctic and 0.03 psu for the Antarctic sea ice is not 'very' high comparing to the observation. For example, the 0.03 psu for the SH is only about half of the 'median' surface snow salinity (0.06 psu) and $\sim$1/30 of the 'mean' snow salinity (=0.9 psu) observed in the Weddell Sea SIZ (see information in Rhodes et al. 2017). It seems to me the overestimation of SSA by the model could be related to one 'missed' process by the model, namely the negative feedback of sublimated water vapour to the ambient air near surface layer, which prevents further evaporation from suspended blown snow particles in the BS layer [Mann et al. 2000]. Thus, it is likely that model (like GOES-chem) without this process could result in overestimated bulk sublimation and then SSA production. I will not blame them not considering this process in their model, as it is out of the range of this study, but it would be useful if some discussions can be made.

Mann, G. W., Anderson, P. S., and Mobbs, S. D.: Profile measurements of blowing snow at Halley, Antarctica, J. Geophys. Res., 105, 24,491–24,508, 2000.

Another factor that could be responsible for the overestimation may come from one assumption made in this model set-up. According to their previous model study (Huang

and Jaegle 2016), they assumed that one wind-blown particles will generate 5 sub-SSA, instead of one as assumed in the original parameterization by Yang et al. (2008). Is this term making some differences? It would be helpful if some discussions can be made as a model sensitivity study.

Yang, X., Pyle, J. A., and Cox, R. A.: Sea salt aerosol production and bromine release: Role of snow on sea ice, Geophys. Res. Lett., 35 (L16815), doi:10.1029/2008gl034536, 2008.

Tiny comments: Figure 1, colour bar needs to be improved. It is hard to distinguish the extinguish coefficient values between ∼10 and ∼20 Mm-1 in the upper panel, and between ∼5 and ∼10 Mm-1 in the bottom panel of figure 1. A similar problem also appeared in other plots. P10 line 4 and Figure 8: longitude/latitude ranges are mentioned, but not shown in in the corresponding plot. Major longitude/latitude information should be given in all relevant figures.

---

## Author Response (AR2)

**Point-by-point responses**

**Response to co-Editor's comments**

In your work, you use the aerosol extinction coefficient as parameter of choice to which model and satellite retrieval are compared to. As you are aware, the ambient extinction coefficient will also depend on the relative humidity due to hygroscopic growth. How are the aerosol optical properties and their hygroscopic growth (e.g. sea spray) parameterized in GEOS-Chem? I think it would be useful to the reader if you add a few lines on this matter to the method section and possibly also to the conclusion and discussion section.

-We have added the following in the revised manuscript:
"The hygroscopic growth of SSA follows parameterization of Lewis and Schwartz (2006). Sedimentation of SSA is calculated throughout the atmospheric column based on the Stokes velocity scheme. Wet deposition of aerosol includes convective updraft, washout and rainout from precipitation (Liu et al., 2001), as well as snow scavenging (Wang et al., 2011).
The aerosol extinction coefficient at 550nm calculated in GEOS-Chem is a function the mass concentrations, extinction efficiency and mass density based on Mie theory, and takes into account the hygroscopic growth of aerosols as described in Martin et al. (2003), with an updated size distribution for SSA (Jaeglé et al., 2011)."

Concerning the CALIOP data: Retrieving extinction coefficients from CALIOP observations requires the assumption of an aerosol-type-specific extinction-to-backscatter lidar ratio, which is another significant source of uncertainty especially for the Arctic region where CALIOP was shown to be generally higher than surface-based in-situ measurements (see Tesche et al., 2014 and references therein). Since this could be another possibility to explain your findings, I would encourage you to add a few more sentence to this aspect to the discussion section.

- Indeed the CALIOP (version 3) tends to overestimates the aerosol extinctions over the Arctic by assigning "polluted dust" lidar ratio to the regions where dust and marine aerosol coexist. This issue is addressed by CALIOP version 4, which assigns a new "dusty marine" regions that has a lidar ratio 33% than "polluted dust" region.
The following discussion is added in the revised manuscripts:

"Extinction-to-backscatter ratios (lidar ratios) are assigned based on the aerosol types for calculations of Level 2 aerosol extinctions (Omar et al, 2009). Tesche et al. (2014) showed that CALIOP v3.01 aerosol extinctions coefficients overestimated in situ observations in the Arctic by 47%. For version 4.10 product, a new subtype "dusty marine" is assigned when dust and marine aerosol coexists, with a lidar ratio 33% smaller than that of polluted dust (Winker, 2016), which brings the in situ and CALIOP extinction coefficients in closer agreement (not shown)."

A few more minor comments:

Figure 6 misses the colorbars in the different panels (a-c and e-h).
-We have added the missing colorbars.

Figure 8: Please improve the colorbars for panel c and d (in the later one, the upper and lower boundaries look are not consistent).
Please add correct labels.

-We have updated the colorbars in Figure 8.

Figure S1 & S2: Please add a label to the colorbar.

-We have added the labels to colorbars in Figure S1 and S2.

**Report #1**
Authors corrected the manuscript on basis of my questions and comments. I can suggest this manuscript for publication. However, I suggest several technical corrections before publication, as follows.

Figure 6:
Legends overlapped on symbols in the plot. Modify the plot size and legends.

-We have adjusted the symbols and legends in Figure 6.

Figure 7:
Labels in x-axis overlapped. Make more space between each label.
Outward ticks in both X- and Y- axis are better to interpret counter plots.

-Fixed.

Figure 8:
Outward ticks in both X- and Y- axis are better to interpret counter plots.

-Added.

Figure S4:
Labels in x-axis overlapped. Make more space between each label.

-Fixed.

We would like to thank all referees and Dr. Yang for their helpful comments and insightful suggestions.

**Reply to comments by Anonymous Referee #1**

This paper describes atmospheric cycles of sea-salt aerosols in polar regions using model and remote sensing measurement (CALIOP). Authors applied and improve the model, GEOS-chem., to simulate spatial distribution and origins of sea-salt aerosols on basis of various parameters such as salinity of surface snow. They derived an interesting conclusion that sea-salt aerosols in the winter were involved in blowing snow rather than frost flowers on sea-ice. On the whole, the topic of the manuscript is relevant and suitable for the scope of the "Atmospheric Chemistry and Physics. The topics and results deserve to be made available to the scientific community and to be exploited in terms of atmospheric aerosols and ice core community in polar regions. Therefore, this study adds very useful information to our knowledge on the sea-salt cycles involved in blowing snow in polar regions. From this reason, I support publication of this work in ACP. However, the current version contains obvious weaknesses, therefore I recommend a major revision. Details are shown as follows.

1. Size distributions of sea-salt aerosols In the GEOS-Chem. Model, spatial distributions of the concentrations of sea-salt aerosols were calculated on assumption of dry deposition velocity and emission from some origins (e.g., open water, frost flowers, and snow). Sea-salt aerosols were distributed from ultrafine to coarse modes in the polar regions during winter – spring (e.g., Hara et al., ACP, 2011).

Hara, K., et al.: Seasonal features of ultrafine particle volatility in the coastal Antarctic troposphere, Atmospheric Chemistry and Physics, doi:10.5194/acp-11-9803-2011, 2011.

What is procedure to calculate and treat size distributions and concentrations of sea- salt aerosols? What are the initial size distributions of particles immediately after emission from sea-ice and ocean? I think that these parameters are probably as same as those in your previous work (Huang and Jaegle, ACP, 2017). If so, add short explanation about processing of aerosol size distribution in the model for readers. If not, details should be mentioned.

Yes, our size distribution assumptions are the same as in Huang and Jaeglé (2017) and we have clarified this in the revised manuscript:

"We use two SSA size bins: accumulation mode (rdry = 0.01−0.5 μm) and coarse mode (rdry = 0.5–8 μm)."

"The size distribution of wind-lifted snow particles follows a two-parameter gamma distribution (Yang et al., 2008 and references therein). Once sublimated, snow particles are released as SSA particles. We assume that 5 SSA particles are produced per snowflake (N=5) based on a comparison against observations of submicron SSA mass concentrations at Barrow, Alaska (Huang and Jaeglé, 2017). The size distribution of blowing snow SSA is determined from the

We would like to thank all referees and Dr. Yang for their helpful comments and insightful suggestions.

**Reply to comments by Anonymous Referee #1**

This paper describes atmospheric cycles of sea-salt aerosols in polar regions using model and remote sensing measurement (CALIOP). Authors applied and improve the model, GEOS-chem., to simulate spatial distribution and origins of sea-salt aerosols on basis of various parameters such as salinity of surface snow. They derived an interesting conclusion that sea-salt aerosols in the winter were involved in blowing snow rather than frost flowers on sea-ice. On the whole, the topic of the manuscript is relevant and suitable for the scope of the "Atmospheric Chemistry and Physics. The topics and results deserve to be made available to the scientific community and to be exploited in terms of atmospheric aerosols and ice core community in polar regions. Therefore, this study adds very useful information to our knowledge on the sea-salt cycles involved in blowing snow in polar regions. From this reason, I support publication of this work in ACP. However, the current version contains obvious weaknesses, therefore I recommend a major revision. Details are shown as follows.

1. Size distributions of sea-salt aerosols In the GEOS-Chem. Model, spatial distributions of the concentrations of sea-salt aerosols were calculated on assumption of dry deposition velocity and emission from some origins (e.g., open water, frost flowers, and snow). Sea-salt aerosols were distributed from ultrafine to coarse modes in the polar regions during winter – spring (e.g., Hara et al., ACP, 2011).

Hara, K., et al.: Seasonal features of ultrafine particle volatility in the coastal Antarctic troposphere, Atmospheric Chemistry and Physics, doi:10.5194/acp-11-9803-2011, 2011.

What is procedure to calculate and treat size distributions and concentrations of sea- salt aerosols? What are the initial size distributions of particles immediately after emission from sea-ice and ocean? I think that these parameters are probably as same as those in your previous work (Huang and Jaegle, ACP, 2017). If so, add short explanation about processing of aerosol size distribution in the model for readers. If not, details should be mentioned.

Yes, our size distribution assumptions are the same as in Huang and Jaeglé (2017) and we have clarified this in the revised manuscript:

"We use two SSA size bins: accumulation mode (rdry = 0.01−0.5 μm) and coarse mode (rdry = 0.5–8 μm)."

"The size distribution of wind-lifted snow particles follows a two-parameter gamma distribution (Yang et al., 2008 and references therein). Once sublimated, snow particles are released as SSA particles. We assume that 5 SSA particles are produced per snowflake (N=5) based on a comparison against observations of submicron SSA mass concentrations at Barrow, Alaska (Huang and Jaeglé, 2017). The size distribution of blowing snow SSA is determined from the

size distribution of snow particles, N (=5), and salinity. The resulting emitted mass of blowing snow SSA is obtained by integrating this size distribution into the two SSA size bins."

"The size distribution of SSA from frost flowers follows a lognormal size distribution with a geometric mean diameter of 0.015 μm and a geometric standard deviation of 1.9 (Xu et al., 2013). This size distribution is integrated into the two GEOS-Chem SSA size bins to obtain the emitted mass of SSA from frost flowers."

2. Dry deposition velocity. In this study, aerosol dry deposition velocity was fixed to 0.03 cm s-1, corresponding to that of particles with size of ca. 2um in diameter. As shown by Rhodes et al. (2017) and Hara et al. (2017), sea-salt aerosols and ice particles containing sea-salts were released from snow and frost flowers on sea-ice. Then, size of sea-salt particles and ice particles containing sea-salts can be changed through sublimation and efficient dry deposition of larger sea-salt particles in the atmosphere. In general, the coarser aerosols have larger dry deposition velocity (shorter residence time). Therefore, processing of initial size distribution and modification of size distribution involved simultaneously with dry deposition and sublimation is the most important to simulate the concentrations and spatial distribution of sea-salt aerosols. Because aerosol dry deposition velocity has size-dependence, the fixed and assumed aerosol dry deposition velocity can result in mis-estimation. I understand that it is difficult to input all parameters in model calculation. However, sensitivity of dry deposition on the sea-salt concentrations should be checked. Ideally, size dependence of dry deposition velocity is included in the model (I do not require it this time, but I hope it for progress in the future).
Rhodes, R., Yang, X., Wolff, E., McConnell, J. and Frey, M.: Sea ice as a source of sea salt aerosol to Greenland ice cores: a model-based study, Atmospheric Chemistry and Physics, 17(15), 9417–9433, doi:10.5194/acp-17-9417-2017, 2017.

Our description of the dry deposition parameterization in GEOS-Chem was unclear and incomplete. The constant 0.03 cm/s dry deposition velocity over snow and ice applies to all aerosols except dust and sea salt. For sea salt in particular, we do assume a size dependent dry deposition velocity. More detail on the sea salt deposition velocity parameterization is now given in the revised manuscript:

 "Dry deposition in the GEOS-Chem follows a standard resistance-in-series scheme based on Wesely (1989) as described by Wang et al. (1998). Dry deposition of SSA in the model follows the Zhang et al. (2001) size-dependent scheme over land, and is calculated based on the Slinn and Slinn (1980) deposition model over ocean and sea ice, as implemented by Jaeglé et al. (2011) in GEOS-Chem. The strong size-dependence of SSA deposition is taken into account by integrating the dry deposition velocity over each of the 2 SSA size bins using a bimodal size distribution including growth as a function of local relative humidity (RH). Sedimentation of SSA is calculated throughout the atmospheric column based on the Stokes velocity scheme."

3. Potential frost flower (PFF) coverage PFF were estimated using air temperature and thickness of sea-ice in the study. Actually, frost flower can be formed on new and young sea-ice. In this study, threshold of newly formed sea-ice thickness is 10 cm. In my experience, this value is

small, because frost flower can be appeared on sea-ice even with thickness of ca. 30cm. If the threshold was smaller, the model results can be underestimated. What is the impact of sea-ice thickness in the model?

The assumed thickness threshold in the model has relatively little impact on our frost flower emissions. Following the suggestion of this reviewer, we conducted a sensitivity study inhibiting the frost flower emission with sea ice thickness over 30cm. We find that frost flower emissions increase by less than 10% over the Arctic, and less than 1% over the Antarctic. This small sensitivity to the assumed the sea ice thickness threshold is a result of the significant decrease in frost flower growth rate with increasing ice thickness (Kaleschke et al., 2004).

Kaleschke, L., Richter, A., Burrows, J., Afe, O., Heygster, G., Notholt, J., Rankin, A. M., Roscoe, H. K., Hollwedel, J., Wagner, T., and Jacobi, H.-W.: Frost flowers on sea ice as a source of sea salt and their influence on tropospheric halogen chemistry, Geophys. Res. Lett., 31, L16114, doi:10.1029/2004GL020655, 2004.

4. Spatial distribution (Figs. 1 and 4) Spatial distribution of aerosol extinction coefficients and model results during cold seasons were depicted in Figs. 1 and 4. These plots provide us very interesting information to understand atmospheric sea-salt cycles in Arctic and Antarctica. However, these periods correspond to develop sea-ice extent, so that these distributions included also seasonal feature of sea-ice extent, which is associated with origins of sea-salt aerosols. Seasonal and spatial variations of source strength and origins of sea-salt aerosols should be taken into account. To exclude influences of the seasonal features, I suggest that the selected months are shown, for example month with maximum of sea-ice extent (March in Arctic and September in Antarctica).

Following the suggestion of this reviewer, we have now added two figures showing monthly mean aerosol extinction coefficients from CALIOP and models during each cold month (November-April in Arctic and May-October in Antarctica) in the Supplementary material (Figures S5 and S6). Overall, the monthly comparisons are consistent with the cold-season comparison, with STD+Opt. Snow best capturing the spatial distributions of CALIOP aerosol extinction coefficients among four model simulations.

**Reply to comments by Anonymous Referee #2**

This is a well written manuscript describing modeling of Arctic aerosol and comparison of these models to observations from CALIOP satellite lidar observations. The model, GEOS-Chem, uses various parameterizations of aerosol production mechanisms, and addition of a blowing snow mechanism brings the model closer to observations. The blowing snow model is further refined by varying the surface snow salinity to improve agreement with observations. An example of an event of blowing snow is shown.

Overall, I feel that this is a well written manuscript, but that the identification of model modifications with specific physical processes sometimes goes further than is justified and/or alternative hypotheses have not been explored fully. The CALIOP data indicate that there is larger extinction present near the surface than the model would indicate, so a wind speed and snow salinity dependent blowing snow model is added, increasing the modeled aerosol extinction, which brings it closer to observations. However, one needs to consider how definitive the identification of these model variables is with physical processes. Specific questions in this regard are:

1) After adding "blowing snow", the model is tuned to reduce surface snow salinity in MYI areas as compared to FYI areas and over the wintertime season. How robust is the necessity to tune down the salinity? For example, Figure 3 shows distributions of extinction in FYI, MYI, and CAA areas. Visually, I can barely see any difference between the CALIOP observations in panels g, h, and i. Values are about 15 Mm^-1 from Jan-Apr, low in summer, and increase back to 15 Mm^-1 towards the end of the year. Is there any statistical difference between these monthly observational distributions? Given the lack of difference between these locations, it seems like the need to optimize the model is weak. Specific monthly values are listed, but it doesn't seem like there is enough information to actually map out this amount of information. For example, could a different single fixed value of salinity be used to optimize the model similarly? It is not unreasonable that surface snow salinity would decrease as you add new snow (which is of low salinity), but the question is how strong the modeling evidence for this decrease is. Please show that the trend from the "optimization" is a real effect larger than statistical errors.

We agree with this reviewer that the need for an optimized time-dependent snow salinity was not very clear in our original manuscript. This was because we made changes with opposing effects partially cancelling each other out, especially for the Arctic: reducing the salinity of MYI and applying a monthly-varying optimized salinity for FYI based on CALIOP extinction observations. As described in the introduction of our manuscript, the lower salinity of MYI and snow on MYI has been clearly demonstrated by observations and is well accepted. In the revised manuscript our STD+Snow simulation now includes this more realistic assumption by lowering the salinity of snow on MYI to 0.01 psu in the Arctic and 0.003 psu in the Antarctic. The new STD+Opt. Snow has the same salinity for MYI as the STD+Snow simulation and uses the fit to CALIOP extinctions to optimize salinity. The impact of the changing snow salinity can be more clearly seen in the revised Figure 3 for FYI (panel g), especially for October-December when the higher salinities on young sea ice (0.36-0.19 psu) lead to a near doubling in aerosol extinction, consistent with CALIOP. We also perform Student's t-test for each cold month and find that significance between these two models' bias are smaller than 0.05 on both FYI and MYI, except

for Antarctic MYI in May, which indicates that STD+Snow and STD+Opt. Snow are statistically significantly different. Applying a single scaling factor for the salinity of surface snow on FYI does not address this model discrepancy over the Arctic. This is now made clearer in the revised manuscript.

"We also examined whether a single fixed value of salinity over FYI can lead to similar improvements in the agreement with CALIOP. The resulting fixed salinities are 0.11 psu over Arctic FYI and 0.018 psu over Antarctic FYI, leading to good overall agreement with CALIOP over the Antarctic (NMB of +5% on FYI and –9% on MYI) with no significant improvement seen in the Arctic (NMB of –7% on FYI and –18% on MYI). We found that over the Arctic, a simulation using a single salinity of 0.11 psu (STD+Const. Snow, Fig. S8g–h) yields results similar to the STD+Snow simulation and cannot explain the high extinction values during fall/early winter. Over Antarctic sea ice, the performance of a simulation with 0.018 psu over FYI shows results similar to the STD+Opt. Snow simulation. Thus there is a stronger case for using a seasonally varying snow salinity over Arctic sea ice than over Antarctic sea ice. We speculate that this might be linked to relatively smaller seasonal variation in sea ice thickness and snow depth for Antarctic sea ice compared to the Arctic. In their snow climatology, Warren et al. (1999) report that the mean snow depth at an Arctic sea ice site increased from 8.7 cm in October to 28.9 cm in March. Satellite-based observations of Arctic FYI thickness show an increase from 0.95 m in October to 2.15 m in May (Kwok and Cunningham, 2015). In contrast, over Antarctic sea ice the mean sea ice thickness and snow depth remained fairly constant during fall–winter (April: 0.48 m for ice thickness and 0.11 m for snow depth; August: 0.52 m for ice thickness and 0.11 m for snow depth) as described in Worby et al. (1998)."

2) Open water areas can produce aerosol directly (by wind blowing over the exposed sea water) or via re-freezing, which might produce frost flowers and/or simply provide a non-snow-covered highly saline surface that snow could blow onto/across. The manuscript does not do justice to hypotheses other than frost flowers. It should leave open the possibility that open water or thin snow cover on ice could be responsible. For instance, the citation below indicates that open water is a source of sea salt aerosol.

May, N. W., P. K. Quinn, S. M. McNamara, and K. A. Pratt (2016), Multiyear study of the dependence of sea salt aerosol on wind speed and sea ice conditions in the coastal Arctic, J. Geophys. Res. Atmos., 121, 9208–9219, doi: 10.1002/2016JD025273.

We now mention this source from leads in the revised manuscript both in the Introduction and in Section 2.3:

"Over polar regions, SSA can also be generated via sublimation of saline blowing snow (Simpson et al., 2007; Yang et al., 2008), wind-blown frost flower crystals (Rankin et al. 2000; Domine et al., 2004; Xu et al. 2013), and by leads in sea ice (Nilsson et al., 2001; May et al, 2016)."

"In this study, we neglect the role of leads as a source of SSA as we found in Huang and Jaeglé (2017) that while this additional source could potentially be important on local scales near leads, overall the regional increase in SSA emissions is less than 10%."

Another aspect that may affect the ability to model either open water areas of frost flowers is the low spatial resolution (2 x 2.5 degree) of sea ice in the model and also the use a weekly product (Page 5, line 30) for sea ice concentration. This low time resolution and linear interpolation could affect the ability of the model to represent the small spatial scale (few km) and temporally transient sea ice lead features.

Point well taken. Indeed very small and temporary features such as leads might not be well resolved in the MERRA sea ice fields (based on the 12.5 km resolution observations from the SSMI instruments on DMSP satellites), which is why we do not consider this source in our manuscript. As noted above, this is now explicitly mentioned in the revised manuscript.

3) The Canadian Archipelago is a region where there is a great deal of land near sea ice. The land can affect the ability of passive microwave satellites to detect sea ice concentrations (called land contamination), and thus could affect the ability to predict frost flower presence. Also, surface winds in the presence significant topography might not be modeled well at these course spatial resolutions. Therefore, I think that there may be a number of factors in this region and caution against overinterpretation. For example, page 9, line 7 indicates a surface snow salinity of 3 psu (nearly 10% of that of sea water) could reconcile differences. Also, it is stated that Alert is near frost-flower producing regions. I think of Alert being in a MYI area, largely surrounded by older sea ice that builds over years. Please cite sources to indicate evidence for Alert (and Neumayer) being in frost-flower producing area.

The MERRA sea ice component is based on observations from Special Sensor Microwave Imager (SSMI) instruments on Defense Meteorological Satellite Program (DMSP) satellites, which have a native resolution of 12.5km x 12.5km. The proximity of costal ocean grid to land, can indeed lead to false ice concentration signals, and the land contamination errors are quantified in Maslanik et al. (1996). The land contamination errors are relatively small in SSMI, compared to Scanning Multichannel Microwave Radiometer (SSMR) due to higher resolutions (Maslanik et al., 1996). In addition, land masks are used to alleviate the impact of land contamination (Maslanik et al., 1996).  Intercomparison of SSMI sea ice concentrations with ship-based observations and independent satellite products are discussed in Kaleschke et al. (2001), Kern et al. (2003) and Andersen et al. (2007). Overall, the SSMI with ARTIST Sea Ice algorithm (ASI) algorithm yields a reasonable representation of sea ice concentrations.

As noted by this reviewer, Alert is in proximity to MYI region. Our frost flower model predicts the most active frost flower emission region to be over the Canadian Artic Archipelago, and Alert is closest to this region among the three Arctic sites. Therefore, Alert may receive large influence from frost flowers compared to other Arctic sites. We have changed the wording in the revised manuscript to clarify this point.

Andersen, S., Tonboe, R., Kaleschke, L., Heygster, G., and Pedersen, L. T.: Intercomparison of passive microwave sea ice concentration retrievals over the high-concentration Arctic sea ice, J. Geophys. Res., 112, C08004, doi:10.1029/2006JC003543, 2007.

Kaleschke, L., Lüpkes, C., Vihma, T., Haarpaintner, J., Bochert, A., Hartmann, J. and Heygster, G.: SSM/I sea ice remote sensing for mesoscale ocean-atmosphere interaction analysis. Canadian Journal of Remote Sensing, 27(5), pp.526-537, 2001.

Kern, S., Kaleschke, L. and Clausi, D.A.: A comparison of two 85-GHz SSM/I ice concentration algorithms with AVHRR and ERS-2 SAR imagery. IEEE Transactions on Geoscience and Remote Sensing, 41(10), pp.2294-2306, 2003.

Maslanik, J.A., Serreze, M.C. and Barry, R.G.: Recent decreases in Arctic summer ice cover and linkages to atmospheric circulation anomalies. Geophysical Research Letters, 23(13), pp.1677-1680, 1996.

Minor comments:

Page 2, line 20. This sentence is somewhat confusing with respect to what surface is being discussed. Is the top of the newly forming first year ice's salinity being discussed? If so, please clarify that this is the ice surface rather than snow.

Yes, this has been clarified in the revised manuscript.

Page 3, line 3. There is no discussion of open water as a sea salt source.

The role of leads has been included in the introduction of the revised manuscript as part of our response to comment #2.

Page 3, line 27. The wording of "aerosol extinctions and the layers beneath" maybe could be improved.

We have changed the wording in the manuscript.

Page 4, line 20. I think it should be "...with a 1-year..."

This was modified in the revised manuscript.

Page 6, line 26. The wording of "reducing the bias" maybe could be improved (the bias became larger, not smaller, but closer in magnitude to zero).

This has been improved in the manuscript.

Overall, I feel that this manuscript argues well for the need to add a wintertime sea salt aerosol source to the Arctic and this source seems to be effectively modeled by a blowing snow model, but that some further refinements of this model may not be appropriately linked to physical processes (e.g. surface snow salinity changes and frost flowers). Those aspects of the manuscript should be further defended by statistical methods or should be written in a more cautious manner, including alternate hypotheses that seem consistent with the data.

**Reply to short comments by Dr. Yang**

General comments: This manuscript reports a GEOS-chem model study of sea-ice soured SSA (from both blowing snow and frost flowers) and their impacts on polar aerosol extinction. Numerous model results via changing various parameters are per- formed and compared to remote sensing (CALIPSO) data. Some results are quite interesting, adding novel information to our knowledge regarding polar SSA production. Authors even derive an 'optimized' seasonal trend of snow salinity. Due to the lack of year around blowing snow and snowpack salinity measurements on polar sea ice surface, we almost know nothing about seasonal variation regarding snow salinity. For this reason, I will treat their modelling-based seasonal snow salinity as a weakness. Instead, I think it highlight an issue which is largely unknown to our knowledge. As we know snow salinity is one of the critical factor that could determine both salt mass loading and their airborne budget (via affecting size spectrum and then lifetime). Therefore, it is a quite important to investigate this parameter in a modelling study, though it needs justification as reviewers pointed out. In general, this is well written manuscript with some interesting results presented. It fits well the scope of the 'Atmospheric Chemistry and Physics' and will benefit relevant communities in sea ice, ice core and boundary layer chemistry. Thus, I support publication of this work in ACP after a revision (see below my specific comments).

Specific comments: The STD+Snow model run overestimates satellite extinction coefficients. Authors attribute this overestimation to 'higher' snow salinity applied in their model. However, I notice that the salinity levels of 0.1 psu for the Arctic and 0.03 psu for the Antarctic sea ice is not 'very' high comparing to the observation. For example, the 0.03 psu for the SH is only about half of the 'median' surface snow salinity (0.06 psu) and ~1/30 of the 'mean' snow salinity (=0.9 psu) observed in the Weddell Sea SIZ (see information in Rhodes et al. 2017). It seems to me the overestimation of SSA by the model could be related to one 'missed' process by the model, namely the negative feedback of sublimated water vapour to the ambient air near surface layer, which prevents further evaporation from suspended blown snow particles in the BS layer [Mann et al. 2000]. Thus, it is likely that model (like GOES-chem) without this process could result in overestimated bulk sublimation and then SSA production. I will not blame them not considering this process in their model, as it is out of the range of this study, but it would be useful if some discussions can be made.

Mann, G. W., Anderson, P. S., and Mobbs, S. D.: Profile measurements of blowing snow at Halley, Antarctica, J. Geophys. Res., 105, 24,491–24,508, 2000.

Another factor that could be responsible for the overestimation may come from one assumption made in this model set-up. According to their previous model study (Huang and Jaegle 2016), they assumed that one wind-blown particles will generate 5 sub- SSA, instead of one as assumed in the original parameterization by Yang et al. (2008). Is this term making some differences? It would be helpful if some discussions can be made as a model sensitivity study.

Yang, X., Pyle, J. A., and Cox, R. A.: Sea salt aerosol production and bromine release: Role of snow on sea ice, Geophys. Res. Lett., 35 (L16815), doi:10.1029/2008gl034536, 2008.

Indeed, our derived seasonal varying surface snow salinity is based on the hypothesis that the discrepancies between CALIOP and the STD+Snow simulation in the magnitude and seasonal cycle of aerosol extinction coefficients are due to our simplifying assumption of a uniform surface snow salinity over sea ice. We cannot rule out the alternative explanations that the overestimate of Antarctic aerosol extinctions is caused by the fact that our simulation does not include the negative feedback of water vapor sublimation or by our assumption of number of particle produced per snowflake (N).

We mention these possibilities in the revised manuscript. To address Dr. Yang's concerns, we have conducted a sensitivity blowing snow simulation assuming N=1. We have included the following discussion in the revised manuscript:

"It is also possible that the discrepancies between observed and modeled aerosol extinction coefficients are due to other factors in the blowing snow parameterization as implemented in GEOS-Chem. For example, our simulation does not include the negative feedback of water vapor sublimation (Mann et al., 2000): as blowing snow particles sublime in unsaturated air, they cause an increase in water vapor and thus cooling of the surrounding air. Both effects lead to an increase in RH near saturation, reducing the sublimation rate. Another underlying assumption is that 5 SSA are produced for each snowflake that sublimes (N=5). We conducted a sensitivity simulation assuming one SSA per snowflake, shown as STD+Snow (N=1) in the supplement (Fig. S7 and S8). This change does not affect the total emissions of blowing snow SSA, but decreases the fraction of SSA in the accumulation mode (see Huang and Jaeglé, 2017). As the extinction efficiency of accumulation mode SSA is larger than that of coarse mode SSA, the assumption of N=1 leads to a 30–50% decrease in modeled extinctions relative to the STD+Snow (N=5) simulation. Overall, this results in improved agreement with CALIOP observations over Antarctic sea ice, but the CALIOP aerosol extinctions are underestimated over the Arctic. Increasing the surface snow salinity over Arctic FYI can address the model discrepancy in aerosol extinction coefficients, but will lead to a factor of 1.5–2 overestimate in SSA mass concentrations."

Tiny comments: Figure 1, colour bar needs to be improved. It is hard to distinguish the extinguish coefficient values between ~10 and ~20 Mm-1 in the upper panel, and between ~5 and ~10 Mm-1 in the bottom panel of figure 1. A similar problem also appeared in other plots. P10 line 4 and Figure 8: longitude/latitude ranges are mentioned, but not shown in in the corresponding plot. Major longitude/latitude information should be given in all relevant figures.

We have changed the colorbar in Fig. 1 and 4. We have also included the major longitudes and latitude in the relevant figures.

[revised manuscript text omitted]

**3 Monthly maps of aerosol extinction coefficients**

Fig. S5 and S6 show 2007-2009 mean monthly maps of the distribution of aerosol extinction coefficients over the Arctic and Antarctic for November through April (Arctic) and May through October (Antarctic). Monthly extinction coefficients from CALIOP are compared to those from the four GEOS-Chem simulations. Also shown in Fig. S5 and S6 are monthly FYI and MYI sea ice coverage. Overall, the monthly comparisons are consistent with the mean cold-season comparison (Fig. 1 and 4 in the main manuscript), with STD+Opt. Snow best capturing the spatial distributions of CALIOP aerosol extinction coefficients among four model simulations

**4 Sensitivity blowing snow simulations**

We conduct two sensitivity simulations: 1) STD+Snow (N=1), which is the same as the STD+Snow simulation but assumes N=1 (number of SSA particle produced per snowflake) instead of N=5; 2) STD+Const. Snow, same as STD+Snow but applying a higher surface snow salinity on FYI over the Arctic (0.11 psu) and lower surface snow salinity on FYI over the Antarctic (0.018 psu). The choice of the FYI salinity was made to minimize the cold-months bias between GEOS-Chem and CALIOP. Results are shown in Fig. S7 for the Arctic and Fig. S8 for the Antarctic. Over the Arctic, the STD+Opt. Snow has best agreement with CALIOP among four blowing snow simulations. Over the Antarctic, the STD+Opt. Snow, STD+Snow (N=1) and STD+Const. Snow display similar model performances.

Figure S1: Vertical profiles of backscatter coefficients for detected aerosol layers (a—b) and aerosol detection frequency (c) in November—April 2007—2009 at 62°-70°N. The daytime backscatter coefficients shown in (b) are scaled ($\beta'_d=\beta_d/1.6$). Daytime profiles are shown in red lines, and nighttime profiles are shown in blue lines. The black lines in a—b indicate the ratios of daytime-to-nighttime backscatter coefficients. The black line in (c) shows the ratios of daytime-to-nighttime aerosol detection frequency ($f_D/f_N$). Shown in (d) is the scatterplot of $f_D/f_N$ ratio as a function of the mean backscatter coefficients. The colors represent the number of points in each bin. Black circles are the median $f_D/f_N$ ratio for the corresponding mean backscatter coefficient. The black line is the linear fit of the black circles.

[Figure]

Figure S2: Same as Figure S1, but for May—October 2007-2009 at 62°–70°S.

[Figure]

**Figure S3: Comparison of seasonal variations of daytime, nighttime and nighttime-equivalent extinction coefficients over the (a) Arctic and (c) Antarctic. Also shown are the vertical profiles of daytime, nighttime and nighttime-equivalent extinction coefficients in the (b) Arctic winter (November–April) and (d) Antarctic winter (May–October).**

[Figure]

**Figure S4: Seasonal variation of monthly sea ice extent ($10^6$ km$^2$) of total sea ice (black lines), first-year sea ice (FYI, orange lines) and multi-year sea ice (MYI, purple lines) over the (a) Arctic and (b) Antarctic.**

[Figure]

**Figure S5: Monthly mean (2007-2009) spatial distribution of mean aerosol extinction coefficients (0–2 km) for November through April over the Arctic observed by (a) CALIOP and calculated with the GEOS-Chem model in (b) a Standard simulation (STD), (c) a simulation including blowing snow SSA emissions (STD+Snow), (d) an optimized blowing snow simulation (STD+Opt. Snow), and (e) a simulation including frost flower SSA emissions (STD+FF). The simulated extinctions are sampled at the time and location of the CALIOP overpasses, and the CALIOP sensitivity threshold is applied. Also shown are the (f) monthly MERRA sea ice coverage, with light gray shading indicating FYI and black shading for MYI.**

[Figure]

**Figure S6: Same as Figure S5, but for Antarctic cold months (May–October)**

[Figure]

**Figure S7: Cold season (November–April) spatial distribution of mean aerosol extinction coefficients (0–2 km) in the 2007-2009 over Arctic observed by (a) CALIOP and calculated with the GEOS-Chem model in (b) a simulation including blowing snow SSA emissions (STD+Snow), (c) an optimized blowing snow simulation (STD+Opt. Snow), (d) same as STD+Snow simulations but with number of particle produced per snowflake N=1 (STD+Snow (N=1)), and (e) same as STD+Snow simulations but applying a single scaling factor for surface snow salinity (STD+Const. Snow). Middle row: Vertical profiles of Arctic cold season mean aerosol extinction coefficients over (d) FYI, (e) MYI and (f) CAA for CALIOP (black dots with horizontal lines indicating standard deviations) and GEOS-Chem model simulations (STD+Snow: red solid lines, STD+Opt. Snow: red dashed lines; STD+Snow (N=1): blue dashed lines; STD+Const. Snow: green dashed lines). Bottom row: Seasonal cycle of 0–2 km monthly aerosol extinction coefficients averaged over (g) FYI, (h) MYI and (f) CAA. CALIOP observations are shown as black circles and vertical lines indicate the interannual standard deviation. The four GEOS-Chem model simulations are also shown (STD+Snow: red solid lines, STD+Opt. Snow: red dashed lines; STD+Snow (N=1): blue dashed lines; STD+Const. Snow: green dashed lines).**

[Figure]

**Figure S8: Same as Figure S7, but for the Antarctic aerosol extinction coefficients during Austral winter (May–October) over FYI (excluding offshore Ross Ice-shelf), MYI and offshore Ross Ice-shelf. As shown in (f), the monthly average aerosol extinction coefficients are not available over FYI during Antarctic summer (January–March) due to the limited FYI extent.**